# Trio-pharmacophore DNA-encoded chemical library for simultaneous selection of fragments and linkers

Meiying Cui[1], Dzung Nguyen [2], Michelle Patino Gaillez[1], Stephan Heiden[2], Weilin Lin[1], Michael Thompson[2], Francesco V. Reddavide[2] ✉, Qinchang Chen [3,4] ✉ & Yixin Zhang [1] ✉

The split-and-pool method has been widely used to synthesize chemical libraries of a large size for early drug discovery, albeit without the possibility of meaningful quality control. In contrast, a self-assembled DNA-encoded chemical library (DEL) allows us to construct an m x n-member library by mixing an m-member and an n-member pre-purified sub-library. Herein, we report a trio-pharmacophore DEL (T-DEL) of m x l x n members through assembling three pre-purified and validated sub-libraries. The middle sub-library is synthesized using DNA-templated synthesis with different reaction mechanisms and designed as a linkage connecting the fragments displayed on the flanking two sub-libraries. Despite assembling three fragments, the resulting compounds do not exceed the up-to-date standard of molecular weight regarding drug-likeness. We demonstrate the utility of T-DEL in linker optimization for known binding fragments against trypsin and carbonic anhydrase II and by de novo selections against matrix metalloprotease-2 and −9.

Discovering small molecular binders against protein targets of interest is important for many biochemical and pharmaceutical research fields. In recent years, DNA-encoded chemical library (DEL) technologies have emerged as a powerful combinatorial method for ligand discovery in industry and academia[1,2]. Using split-and-pool synthesis, DELs of extraordinarily large size can be synthesized, with the stepwise synthesis of each compound barcoded in the attached DNA sequence[3–7]. By using next-generation sequencing (NGS), the identities and enrichment of selected compounds can be revealed. However, these single-pharmacophore DELs (Fig. 1a), in which each DNA molecule displays one compound, have their drawbacks. As the millions and billions of different DNA-encoded compounds cannot be individually purified and characterized, the purity of a DEL decreases with the increase of reaction steps, and the final quality cannot be controlled[7].

Another combinatorial method for ligand discovery is the fragment-based approach. Fragment-based drug discovery (FBDD) identifies low-molecular-weight ligands that bind to different sites on a

target protein. The structural information regarding the binding mode of these fragments is commonly determined by X-ray crystallography[8,9] or NMR spectroscopy[9,10] and is then used to design linked fragments as potent binders with drug-like properties. A variation of DEL, the self-assembled DEL, also known as dual-pharmacophore DEL (Fig. 1a), displays two compounds at the 3' and 5' ends of a DNA duplex. It resembles the FBDD approach and can facilitate the discovery of low-molecular-weight fragments. Recently, code-transferring methods between two DNA strands have been developed to reveal pairing information of the enriched fragments[11–17]. Dynamic DELs[14,15,18–22] and photo-crosslinking DELs[14,19,23–28] have also been reported to improve the signal-to-noise ratio of selection processes and to allow in-solution DEL selections, respectively. Dual-pharmacophore DEL has the advantage of constructing large libraries with high purity. For example, purifying every compound in a single-pharmacophore library with 1 million members is impractical. However, assembling two 1000-compound-DNA-encoded sub-libraries can

[1]B CUBE, Center for Molecular Bioengineering, Technische Universität Dresden, Dresden, Germany. [2]DyNAbind GmbH, Dresden, Germany. [3]Research Institute of Intelligent Computing, Zhejiang Lab, Hangzhou, China. [4]School of Life Sciences and Technology, Tongji University, Shanghai, China. ✉e-mail: francesco.reddavide@dynabind.com; chenqc@zhejianglab.edu.cn; yixin.zhang1@tu-dresden.de

also result in a library with 1 million members, and every DNA-encoded compound can be purified by high-pressure liquid chromatography (HPLC) and characterized by mass spectrometry. However, dual-pharmacophore DELs also share the drawback of other FBDD methods, as discovering an optimal linkage between two fragments is always time-consuming and labor-intensive.

Melkko, Scheuermann, et al. postulated a triplex DEL in 2004[29,30] (Fig. 1a). It would result in larger self-assembled DELs, in which every member can be purified and characterized. However, the construction of the triplex DEL has not been reported, as the challenges are not only associated with the library synthesis. The difficulty in finding optimal linkage between two fragments has made it intimidating to develop a general strategy to assemble three fragments with a repertoire of multi-functional scaffolds. Moreover, the resulting compounds will largely exceed the common criteria regarding drug-likeness on the aspect of molecular weight, e.g., the Lipinski rule or the up-to-date molecular mass cut-off based on the properties of orally available small molecules approved in the past decade[31].

In this work, we design a trio-pharmacophore DEL (T-DEL), in which sub-library B (SL-B) is used as a scaffold to assemble the other two sub-libraries (SL-A and SL-C) (Fig. 1b). The SL-B cannot only mediate the distances between the fragments in SL-A and SL-C but also introduce additional contacts with the protein. DNA-templated synthesis (DTS) is used to synthesize the SL-B (Fig. 1c). In conventional DEL syntheses using DTS, the organic compounds are detached from one DNA strand, presented, and encoded on the other DNA strand in the final construct[32–38]. For the SL-B of T-DEL, the small molecular compounds are flanked by two DNA strands, which are used to assemble the SL-A and SL-C. Using this design, we synthesize a T-DEL with over 20 million members, in which every conjugate is purified via polyacrylamide gel electrophoresis (PAGE) or HPLC and characterized by mass spectrometry. After selection, the fragments revealed from SL-A and SL-C can be connected by the selected linker fragments from SL-B, resulting in potent small molecular binders against the protein target of interest.

## Results
### Library design and synthesis
The synthetic route of sub-library B (SL-B) is shown in Fig. 1c. We designed two 33-nt oligonucleotides that were partially complementary with 6 plus 12 base pairs. Various bi-functional building blocks were conjugated to the oligonucleotides at the 3' or 5' termini, resulting in 3' and 5' conjugates, respectively (Supplementary Fig. 1). Then, the conjugates underwent different DNA-templated reactions between matching functional groups to generate DNA-compound-DNA conjugates. The reactions were monitored by denaturing urea PAGE, and the reaction products were purified from the gel. The molecular weight of the reaction products was confirmed by LC-ESI-MS (Fig. 1d and Supplementary Note 1). By employing a variety of reaction types, such as amide bond formation, reductive amination, azide-alkyne cycloaddition, Michael addition, and Diels-Alder reaction, we have generated 30 conjugates covering four structural categories (Supplementary Fig. 2). Each DNA-compound-DNA conjugate was then encoded by splint ligation. The encoding process was also monitored by denaturing PAGE, and only the successfully encoded conjugate was purified from the gel to ensure the high purity of the library members of SL-B (Fig. 1d).

An 883-member fragment sub-library (SL-A) and an 890-member fragment sub-library (SL-C) were synthesized to form a dynamic dual-pharmacophore DEL[15,18], which are partially complementary by 6 bp. SL-A and SL-B share a 33 bp complementary region, and SL-B and SL-C share a 13 bp complementary region. We examined whether the SL-B can assemble with the SL-A and SL-C to form a stable T-DEL using native DNA PAGE. As shown in Fig. 1e, when SL-A and SL-B (lane 4) or SL-B and SL-C (lane 6) were mixed and allowed to anneal, the bands

indicative of the assembled duplexes were observed. As expected, the mixture of SL-A and SL-C did not form a larger complex (lane 5). When all three sub-libraries were mixed and allowed to anneal (lane 7), the highest band corresponding to the assembled trimeric complex was observed.

### T-DEL for linker optimization
To investigate the use of T-DEL to optimize linkage between fragment pairs, we performed affinity maturation selections against the model proteins bovine carbonic anhydrase II (CAII) and bovine trypsin with their known ligand pairs. As depicted in Supplementary Fig. 3a, we utilized the reported fragment pair of CAII[15], aryl sulfonamide, and 3-{5-[3-(trifluoromethyl) phenyl]−2-furyl} acrylic acid (compound A) as single-member SL-A and SL-C, respectively. After assembling with the 30-member SL-B, the T-DEL library was selected against CAII immobilized on solid support (Fig. 2a, target selection). Selection against blank solid support served as a negative control. Selection with SL-B assembled with non-modified SL-A and SL-C was also performed (Supplementary Fig. 3a, no-ligand target selection). qPCR was used to quantify the amount of each member of SL-B in the three selections with code-specific primers (Supplementary Note 3 and Supplementary Fig. 54). In Fig. 2a, the enrichment was calculated by normalizing the enrichment profile against no-target selection. As expected, the enrichment of the entire SL-B was higher in the target selection than in the no-ligand target selection, demonstrating that the ligand pair facilitates the interaction of SL-B members with the target.

We have chosen two compounds with the highest enrichment (c1 and c2), two with moderate enrichment (c3 and c4), and one compound, c5, with low enrichment for further off-DNA synthesis and validation. These selected structures from SL-B were used to connect sulfanilamide and compound A, resulting in compounds C-1 to C-5 (Fig. 2b and Supplementary Note 2). We also synthesized compound C-0 by connecting sulfanilamide with compound A without a linker moiety. The compounds C-0 to C-5 were validated in an enzyme inhibition assay to measure the $IC_{50}$ values. Sulfanilamide showed an $IC_{50}$ value of 13.36 μM, and compound A exhibited moderate inhibition at 100 μM (Fig. 2c). The compound with the highest enrichment (C-2) displayed a 20-fold improvement in the $IC_{50}$ value (0.67 μM). Compounds with moderate and low enrichment, C-3, C-4, and C-5, exhibited lower inhibitory effects than C1 and C2, agreeing with the selection outcome. Interestingly, C-0 showed the second-highest inhibitory effect ($IC_{50}$ 0.83 μM).

We implemented molecular docking studies to gain more insights into the compounds' binding mechanism and compared the docking poses among the compounds (Fig. 2, Supplementary Figs. S4, S5, and Supplementary Discussion 1). As reported previously[39,40], the sulfonamide moiety binds deeply in the catalytic site via coordinating with $Zn^{2+}$, and forming two hydrogen bonds with Thr198, and one with Pro200 (Supplementary Fig. 4b). Conjugation of compound A to sulfanilamide contributed predominantly to the hydrophobic interactions with the protein, as shown in Supplementary Fig. 4a. The sulfanilamide moiety remained well-positioned in the active site in all re-synthesized compounds (C-0 to C-5) (Fig. 2 and supplementary Fig. S4). We then investigated the docking pose of each compound to understand the different inhibitory effects associated with the linker moieties. The binding pose of C-0 resembled the ligand in the reported crystal structure (PDB:" 6SKV") (Fig. 2d and Supplementary Fig. 5). C-2 adopted a compact conformation in the catalytic pocket, forming five hydrogen bonds with the surrounding residues (Fig. 2e). Also, the large hydrophobic effect and low binding energy may support the highest inhibitory effect of C-2 (Supplementary Fig. 4a). On the contrary, the linker moiety of C-3 and C-4 protruded out of the catalytic pocket (Fig. 2f), which may explain their lower inhibitory effects.

Next, we tested the use of T-DEL for linker optimization with bovine trypsin and its ligand pair, 4-aminomethyl benzamidine and

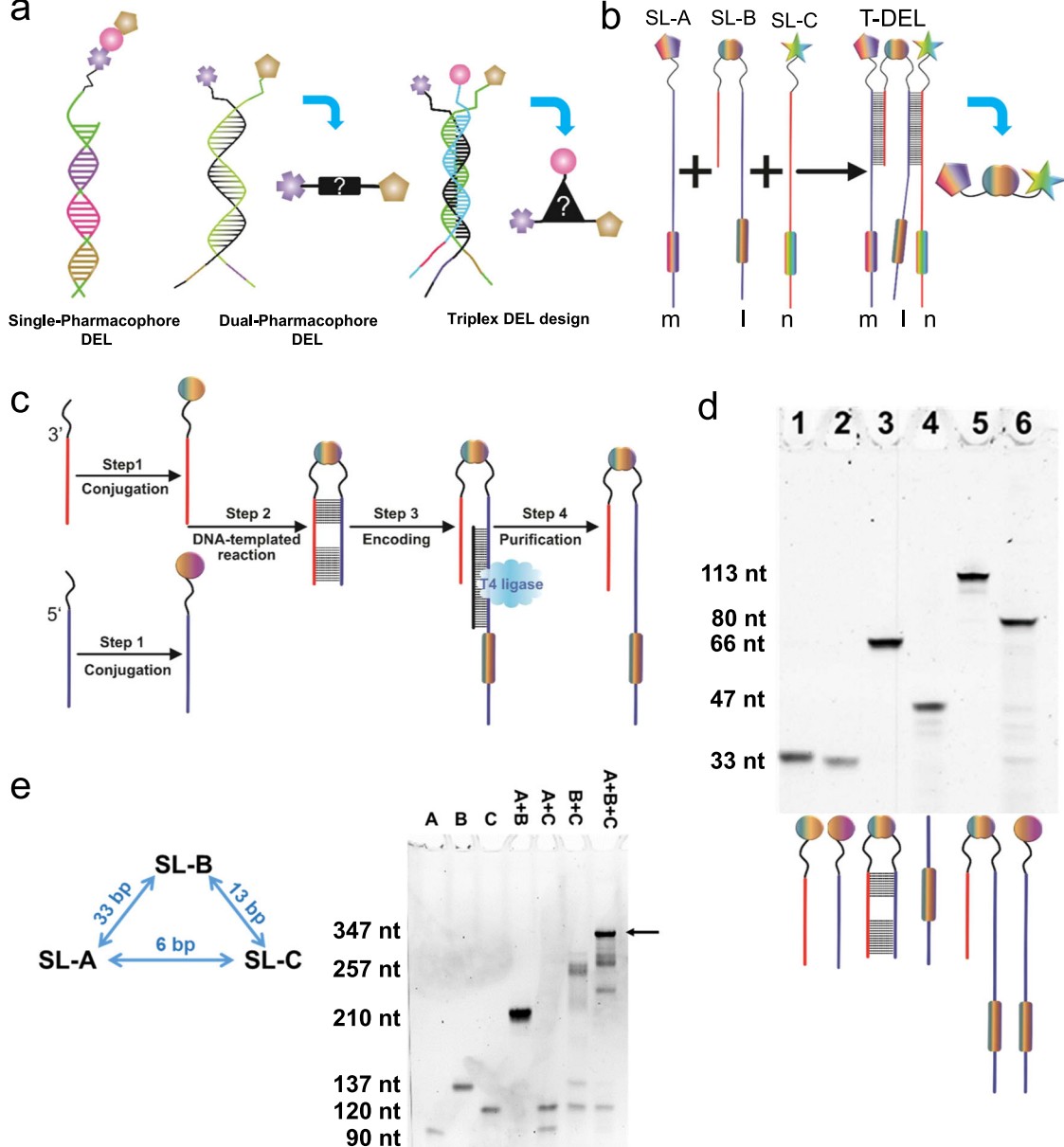

**Fig. 1 | Trio-pharmacophore DNA-encoded chemical library. a** Schematic representation of a single-pharmacophore, a dual-pharmacophore, and a triplex DEL design proposed by Melkko et al. Figure adapted from Melkko et al. 2004 and 2007[29,30]. **b** Format of the trio-pharmacophore DEL (T-DEL) proposed in this work. **c** Synthetic scheme of sub-library B. The synthesis started from two oligonucleotides with functional groups at the 3' and 5' end. The two oligonucleotides shared 6 + 12 complementary base pairs. Step 1. Bi-functional building blocks were conjugated to the oligonucleotides to generate single-side compound-DNA conjugate. Step 2. Conjugates from each side underwent DNA-templated reactions via the complementarity and the reactive functional groups from the building blocks. DNA-compound-DNA conjugates harboring both oligonucleotides were synthesized. Step 3. Each conjugate was then encoded with a unique barcode by using an adapter DNA (in black) and T4 DNA ligase. Step 4. The encoded conjugate was purified from the ligation solution to generate high purity library member of sub-library B. **d** Step-by-step monitoring of the library synthesis by urea-denaturing polyacrylamide gel electrophoresis (Urea PAGE). Lane1 and lane 2 were the single-side conjugates (33 nt). Lane 3 was the DNA-templated reaction product, lane 4 was the barcode DNA (47 nt), and lane 5 was the encoded library member. Lane 6 was the control encoding product of the single-side oligonucleotide (lane 2) and the barcode DNA (lane 4), whose size should be smaller than the encoded conjugate (lane 6) and larger than the conjugate without the barcode (lane 3). The gel was stained by SyBr Green II. The construct of the DNA corresponding to each band is shown under the gel. The experiment was repeated three times. **e** Assembly of three sub-libraries of the trio-pharmacophore DNA-encoded chemical library. Native DNA PAGE was used to monitor the assembly of sub-libraries. Lane1-3: Sub-libraries A, B, and C were loaded separately. Lane 4: Mixture of sub-library A and B. Lane 5: Mixture of sub-libraries A and C. Lane 6: Mixture of sub-libraries B and C. Lane 7: Mixture of all three sub-libraries. The experiment was repeated three times. The DNA sequences are in Supplementary Note 3.

2-iodophenyl isothiocyanate (compound B), reported by the Neri group in their DEL selection with a dual-pharmacophore library[41]. The selection and decoding strategies are identical to CAII (Supplementary Fig. 3b). We have chosen the four highly enriched linker fragments, t1 to t4, and one with low enrichment, t5 (Fig. 3a). The linkers were used to tether the fragment pair to generate small molecules T-1 to T-5 (Fig. 3b and Supplementary Note 2). Again, the two fragments were directly conjugated without a linker, resulting in compound T-0. The compounds were evaluated by an enzyme inhibition assay. 4-aminomethyl benzamidine showed an $IC_{50}$ value of 147.23 µM, in

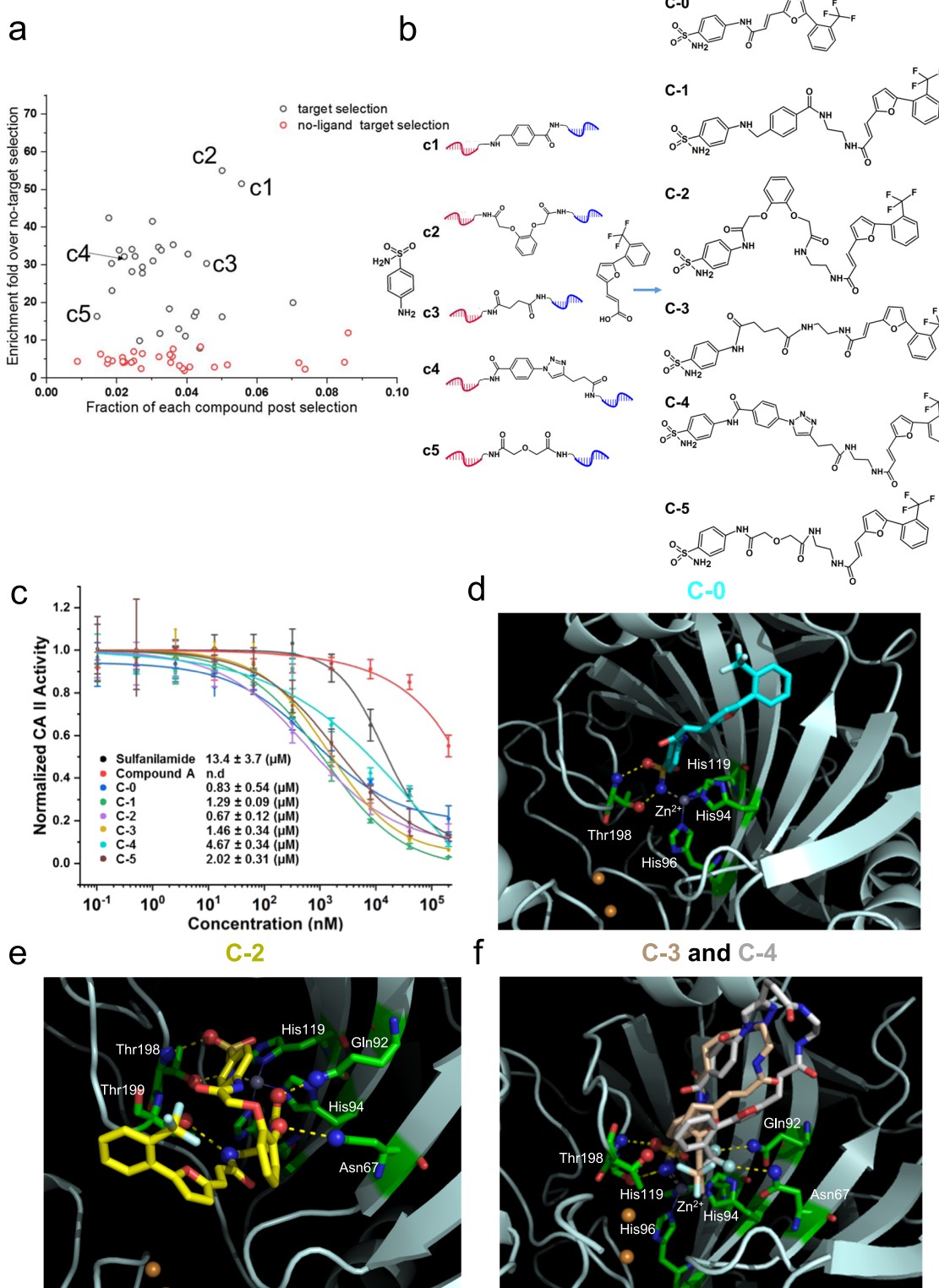

**Fig. 2 | Affinity maturation of ligand pair against bovine carbonic anhydrase II (CAII). a** Scatter plot of the selection outcome. **b** Five conjugates (c1 to c5) enriched from the selection with varying enrichment folds were chosen to link sulfanilamide and compound A, generating small molecules (C-1 to C-5) containing three moieties from the three sub-libraries. C-0 is a direct conjugation between sulfanilamide and compound A. **c** Hit validation by an enzyme inhibition assay. Data are presented as mean values ± SD from three independent measurements. $IC_{50}$ values are presented as mean values ± SE. Data are presented as mean values ± SD from three

independent measurements ($n = 3$ biological replicates). n.d: not detected **d–f** Docking poses of C-0, C-2, C-3, and C-4 in complex with CAII (PDB ID: "6SKV"), accordingly. The protein is in cartoon style, certain residues, and the compounds are in stick representation, and the hydrogen bond-forming atoms are in ball representation. Yellow dashed lines indicate hydrogen bonds, and gray dashed lines stand for the coordination with $Zn^{2+}$. The gray sphere represents $Zn^{2+}$, and the orange spheres represent $Cu^{2+}$. Source data are provided as a Source Data file.

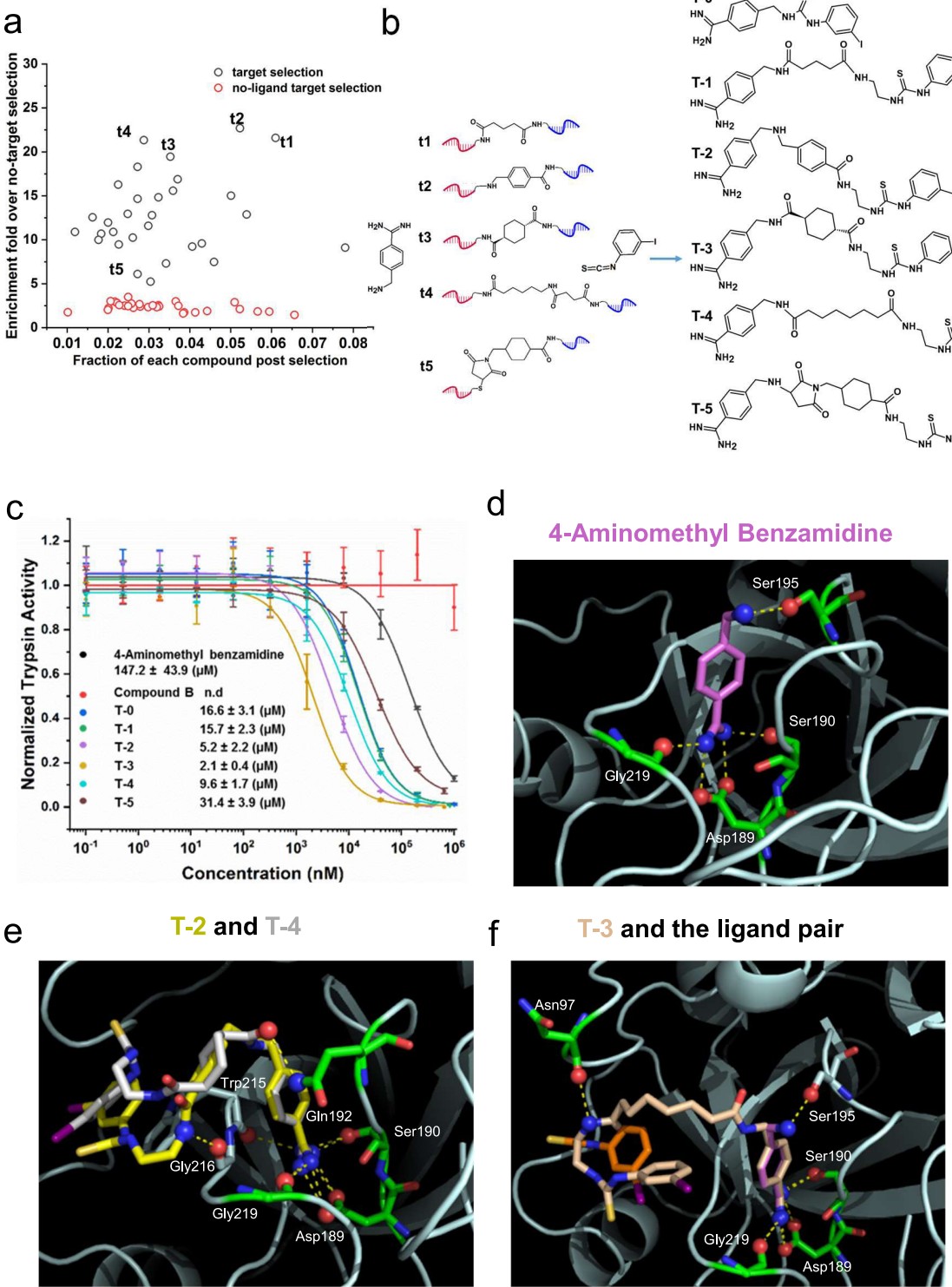

**Fig. 3 | Affinity maturation of ligand pair against bovine trypsin. a** Scatter plot of the selection result. **b** Five conjugates (t1 to t5) enriched from the selection with varying enrichment folds were chosen to link 4-aminomethyl benzamidine and compound B, generating small molecules (T-1 to T-5) containing three moieties from the three sub-libraries. T-0 is a direct conjugation between 4-aminomethyl benzamidine and compound B. **c** Hit validation by an enzyme inhibition assay. Data are presented as mean values ± SD from three independent measurements. IC$_{50}$ values are presented as mean values ± SE. Data are presented as mean values ± SD

from three independent measurements (*n* = 3 three biological replicates). n.d: not detected **d–f** Docking pose of 4-aminomethyl benzamidine, superimposed poses of T-2 and T-4, and the docking pose of T-3 and the ligand pair in complex with bovine trypsin (PDB: " 1BTY"). Compound B is orange, and 4-aminomethyl benzamidine is magenta. The protein is in cartoon style, certain residues, and the compounds are in stick representation, and the hydrogen bond-forming atoms are in ball representation. Yellow dashed lines indicate hydrogen bonds. Source data are provided as a Source Data file.

agreement with the previous report[41], while compound B alone did not display any detectable inhibitory effect. T-1 to T-4 showed remarkable enhancement in the inhibitory effect, especially T-2 and T-3, displaying 70-fold and 30-fold improvement, respectively. T-0 showed an approximately 9-fold improvement (Fig. 3c).

We further studied the binding modes of compounds to trypsin by molecular docking (Fig. 3, Supplementary Figs. S6, S7, and Supplementary Discussion 2). As previously reported[42–44], 4-aminomethyl benzamidine binds to the substrate recognition site and forms hydrogen bonds with the key residue Asp189, the neighboring Ser190, Gly219, and Ser195 (Fig. 3d). In analogy to CAII, conjugating compound B to 4-aminomethyl benzamidine by linkers largely increased the hydrophobic contacts with trypsin, maintaining the binding mode of the benzamidine moiety (Supplementary Fig. 6). T-2 and T-4 displayed similarity in terms of the binding site and pose of both fragments, agreeing with their observed inhibitory effects (Fig. 3, Supplementary Figs. 6a and 7a). Notably, T-3 preserved the binding pose of the single compound B best in all conjugates (Fig. 3 and Supplementary Fig. 7a), and T-5 displayed the worst docking score compared to other re-synthesized small molecules (Supplementary Fig. 6a).

Together, the affinity maturation selections have demonstrated the capability of T-DEL to guide linker optimization for known fragment pairs.

T-DEL is an extension of dual-pharmacophore DEL by assembling pre-purified sub-libraries, which led us to the question: how is the performance of T-DEL in delivering potent ligands to the target protein compared to a dual-pharmacophore format? To answer this question, we measured the recovery of the same ligands from selections using both library formats. For this purpose, we utilized three model targets (CAII, trypsin, and alpha-1-acid glycoprotein[11,45]) and their well-characterized ligand pairs (Supplementary Discussion 3 and Supplementary Figs. S8-S12). By comparing one ligand pair in dual-pharmacophore format to the mixture of 30 different combinations in T-DEL format, we have concluded that most members of the linker library (SL-B) do not improve the binding. As the signals from T-DEL represent the average of 30 different combinations, and due to the large difference among different SL-B members in enrichment (Figs. 2a, 3a, Supplementary Fig. 11c, d), only a few members from SL-B can improve the binding. Interestingly, we have observed that with the increase of binding affinity of ligands in SL-A and SL-C, the overall contributions from SL-B on binding can be augmented, as shown by the ligand-dependent enhancement of recovery (Supplementary Fig. 11a, b) in the T-DEL format.

### T-DEL for de novo selections

A 23.576 million-member T-DEL (883 × 30 × 890) was constructed to test its utility in de novo selections. Matrix metalloproteinases (MMPs) are zinc-dependent endopeptidases capable of degrading and remodeling extracellular matrix components[46,47]. They are attractive therapeutic targets as high expression levels were detected in various diseases, such as inflammatory diseases, and at different stages of cancers, including metastasis, invasion, and angiogenesis[47–53]. As shown in Supplementary Fig. 3c, we performed selections against the two gelatinases (human MMP-2 and human MMP-9) to identify binding fragments for later design and synthesis of small molecule inhibitors. After selection, the three sub-libraries were decoded, and the enrichment was calculated by dividing the post-selection fraction (count/total counts) by the pre-selection fraction (Fig. 4a and Supplementary Fig. 13). Selections using a dynamic dual-pharmacophore DEL with the same members (883;× 890) were also performed against MMP-2 and MMP-9 to select relevant hits that enrich through different DEL formats (Supplementary Figs. 3d and 13). Remarkably, we identified common hits using both formats, indicating that these fragments can be specifically enriched independent from the library design, making it more confident for us to consider them as true positive hits.

The enrichment profiles of all three sub-libraries have shown similarities between MMP-2 and MMP-9, presumably due to the high structural homology of the two proteins[54,55] (Fig. 4b). To validate the selection outcome, we chose three fragments from SL-A (66, 182, and 693), three fragments from SL-C (787, 826, and 828), and three linker fragments with the highest enrichment (12, 24, and 10), and two linker fragments with moderate enrichment (1 and 4) from SL-B for further off-DNA synthesis (Fig. 4). We deployed enzyme inhibition assays of MMP-2 and MMP-9 using a fluorogenic peptide substrate for hit validation. We first measured the inhibition of two enzymes by the fragments. As shown in Fig. 5a, b, fragment 182 exhibited the highest inhibitory effect with $IC_{50}$ values of 95.8 μM and 48.1 μM against MMP-2 and MMP-9, respectively. Fragments 693 and 828 displayed $IC_{50}$ values in the high μM range against both targets, while fragment 787 showed a high μM $IC_{50}$ value only against MMP-9. No inhibitory effects could be measured for fragments 66 and 826.

Since the three sub-libraries were independently decoded and analyzed, in order to identify the best combination of the selected fragments, we synthesized 45 (3x5x3) small molecules covering all possible combinations (Supplementary Fig. 14 and Supplementary Note 2). Figure 5c shows the MW distribution of the 45 compounds. They comply with the current drug-likeness criteria regarding MW, e.g., showing an average MW of 503 Da and 90th percentile of 606 Da, similar to the analysis of all approved drug molecules in the past 20 years[31]. We then assayed the compounds against MMP-2 and MMP-9 (Supplementary Figs. 15-17), and the resulting $IC_{50}$ values are shown in Fig. 5d, e. The compounds are grouped by fragment combinations, and each group has five compounds differing by the linker fragments. Like the enrichment profiles (Fig. 4b), the inhibitory effects of the small molecules showed similar patterns on MMP-2 and MMP-9. For both enzymes, the combinations 182 + 828, 182 + 787, 693 + 828, and 693 + 787 displayed higher inhibitory effects than the other pairs, suggesting a synergistic effect from the combinations of these fragments. 66 and 826, showing the weakest inhibition as fragments, resulted in weak binders after connecting them with various linkers. In addition to the fragment pairing, the linking moiety also impacts the inhibitory effect. When the fragments from SL-A and SL-C were linked by the linker fragments enriched from selections (12, 24, and 10), they often showed lower $IC_{50}$ values than those with the controls (1 and 4). Compound 693_12_828 displayed $IC_{50}$ values of 10 μM and 15 μM against MMP-2 and MMP-9, respectively, tens-of-fold improvements compared to the two fragments. Compounds 182_12_828 and 182_24_828 also exhibited enhanced inhibitory effects compared to the starting fragments (Fig. 5f).

Since the fragment combinations 182 + 828 and 693 + 828 displayed higher potency than the other combinations, we applied molecular docking to shed light on their binding modes. The study suggested that the compounds bind to the catalytic domain, accommodating the hydrophobic subsite 1′ (S1′) of the substrate-binding cleft in both enzymes via fragment 182 or 693. Moreover, the 828 moiety displayed interaction with the catalytic $Zn^{2+}$ in all compounds (Supplementary Discussion 4 and Supplementary Figs. 18–22).

## Discussion

In this work, we have realized the synthesis of a trio-pharmacophore DNA-encoded chemical library (T-DEL). The resulting library has the following features:

I.   Every member used in constructing the 23.576 million-member library has been purified and characterized. To our knowledge, this is the largest self-assembled DEL with purified building blocks.

II.  As the SL-B can serve as both a binding and linking fragment and allows us to obtain additional information about the constructive binding moiety to bridge the two flanking fragments. The

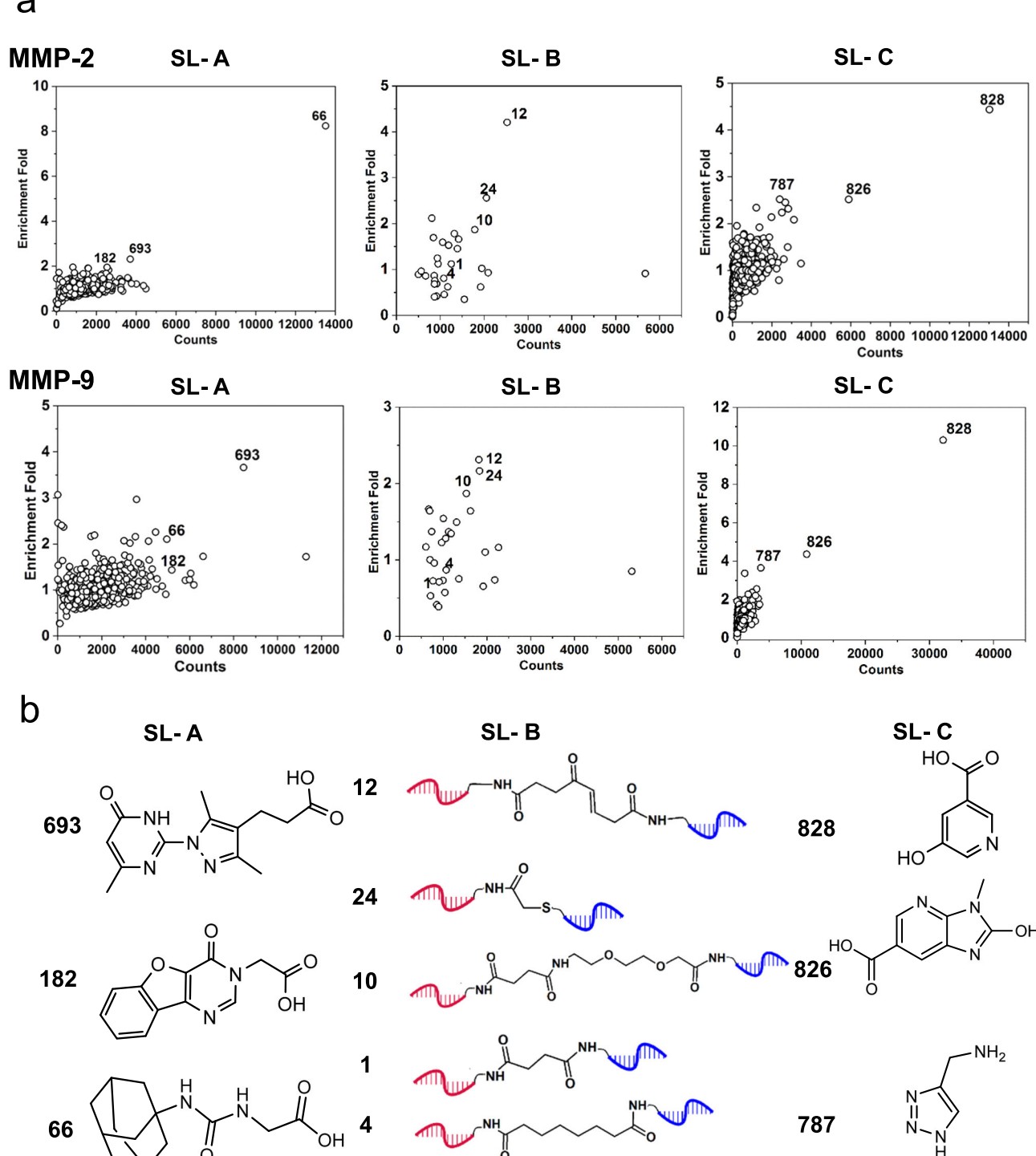

**Fig. 4 | The outcome of the selections against MMP-2 and MMP-9. a** Scatter plots of selection results from sub-libraries A, B, and C. **b** Structures of the selected building blocks from the three sub-libraries for the follow-up hit resynthesis and validation. Fragments **693**, **182**, and **66** were enriched from the SL-A, and fragments **828**, **826**, and **787** were enriched from SL-C. In SL-C, compound **787** was linked to DNA via a succinic acid linker. The linker fragments **12**, **24**, and **10** were enriched from SL-B, and **1**, and **4** were chosen to serve as negative controls. Source data are provided as a Source Data file.

information can guide us in designing a full linker, which is not feasible by dual-pharmacophore libraries.

III. With the T-DEL format, it is possible to create a focused library joining only known fragment pairs, as described for CAII and trypsin, to gain insights into linking the fragments. On the other hand, it is also possible to take full advantage of the chemical diversity and explore all the possible binding modalities, as described for the de novo selections against MMP-2 and MMP-9. However, the inhibitory effect of the re-synthesized compounds is not compelling and needs further optimization.

In summary, the T-DEL strategy has allowed us to optimize the linkers for known fragment pairs and synthesize large DEL for de novo identification of fragments and their linking moieties against protein

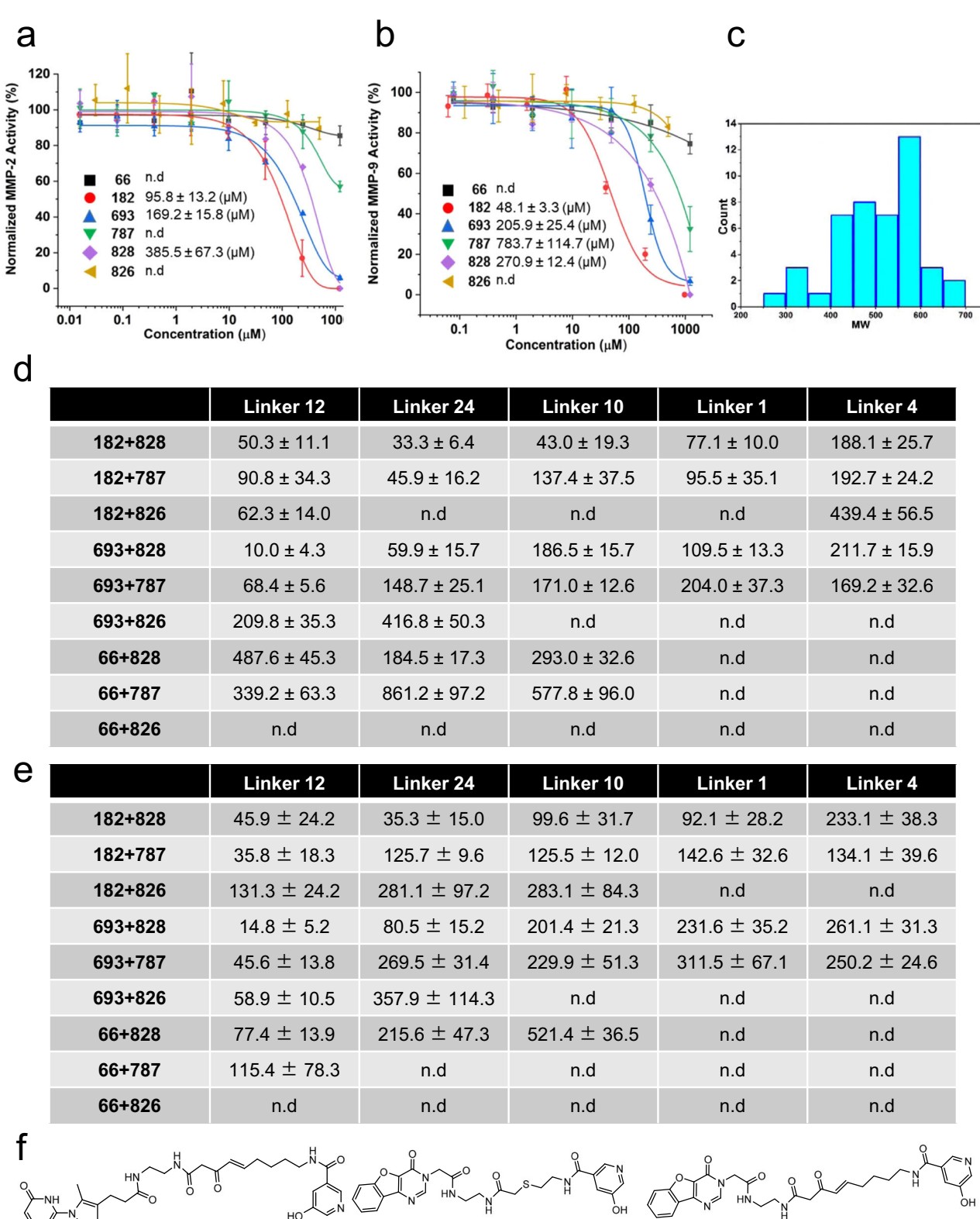

**Fig. 5 | Hit resynthesis and validation. a, b** Fragments enriched from SL-A (**66, 182,** and **693**) and SL-C (**787, 828,** and **826**) were independently validated by enzyme inhibition assay against MMP-2 and MMP-9. Data are presented as mean values ± SD from three independent measurements ($n = 3$ three biological replicates). **c** Histogram of MW distribution of the 45 re-synthesized small molecule compounds. **d, e** IC$_{50}$ values calculated from dose-response measurements (Supplementary Figs. 15, 16, and 17) against MMP-2 and MMP-9 of all compounds. The error bar represents the standard error of the calculated IC$_{50}$. The compounds are grouped by nine fragment combinations. Each group contains five compounds differing by the linker moieties selected from SL-B. Data are presented as mean values ± SE. n.d: not detected **f** The structure of compounds with the best inhibitory effect out of the 45 small molecules. Source data are provided as a Source Data file.

targets of interest. In addition, molecular docking studies revealed the potential binding mode of fragment pairs tethered by different linkers. In the current work, we devised enzyme inhibition assays to validate the hits from affinity selections. It's worth noting the disconnect between the two different assays. As binding does not necessarily exert inhibition, biophysical studies to evaluate the binding affinity shall be performed in the future, to provide information complementary to the in silico study.

A limitation of the T-DEL design is the lack of a code-joining mechanism among the three sub-libraries. Code-transferring[11,14] and code-joining methods[12,13,15–17] between two sub-libraries in dual-pharmacophore DEL can make the pairing information readily available by sequencing the joined codes. In the future, code-transferring and code-joining methods will be investigated to develop an optimal strategy with low interference in library synthesis and selection. It will drastically reduce the cumbersome work of testing different fragment combinations. Further, increasing the size of the linker fragment library (SL-B) is of particular interest via diversifying the scaffold structures by exploring the toolbox of DNA-templated synthesis.

## Methods

### Monitoring DNA-templated reactions by denaturing TBE-urea polyacrylamide gel electrophoresis (PAGE)

A 50% stock gel solution was prepared by dissolving 23.68 g (333 mmol) of acrylamide and 1.32 g (8.56 mmol) $N$, $N'$-methylenbisacrylamide in 50 ml MilliQ water.

Composition of 5× Tris/Borate/EDTA (TBE) buffer (1 L): Tris-base (54 g), boric acid (27.5 g), and 20 ml of 0.5 M ethylenediaminetetraacetic acid (EDTA) (pH 8.0).

Composition of the 10% TBE-urea gel: 1.6 ml of 5× TBE buffer, 1.6 ml of 50% stock gel solution, 3.84 g urea (8 M), 40 μl 10% (m/v) ammonium persulfate, 4 μl tetramethyl ethylenediamine (TEMED), and water up to 8 ml.

The mixed solution was added to a mini gel cassette (Thermo Fisher Scientific, Waltham, USA) and left still for 20 min for polymerization. After polymerization, the cassette was loaded in a gel electrophoresis chamber (Bio-Rad Laboratories, Hercules, USA). 1× TBE buffer was added to the inner and outer chambers. Before loading samples, each lane was flushed with buffer to remove excess urea precipitated during polymerization. DNA samples were mixed with 2× TBE-urea loading buffer (Thermo Fisher Scientific, Waltham, USA) in 1:1 volume to each lane.

The running was performed under a constant voltage of 90 V for 3 h, till the loading dye reached the end of the gel. Next, the gel was detached from the cassette, stained with 1× SYBR Green II for 15 min, and imaged with ChemiDoc MP (Bio-Rad Laboratories, Hercules, USA).

### Ethanol precipitation

0.1 volume of 3 M NaOAc, pH 4.7 was added to the DNA solution. 3.5-fold volume of cold absolute ethanol was added and allowed precipitation at −20 °C o/n. The solution was then centrifuged at 20,784 × $g$ for 30 min at 4 °C and the supernatant was discarded. The pellet was rinsed with 1 ml of 70% cold ethanol by strong vortexing and subjected to a second centrifugation at 20,784 × g for 30 min at 4 °C. The supernatant was discarded and the pellet was dried by a vacuum concentrator. The recovered sample was dissolved in water and quantified by measuring UV absorption (260 nm).

### Encoding of DNA-compound-DNA conjugates

Individual DNA-compound-DNA conjugate purified from denaturing urea PAGE was encoded with a unique DNA sequence using T4 ligase. Each code sequence was phosphorylated at the 5′ terminal by T4 Polynucleotide Kinase according to the manual. The phosphorylated code was used for encoding without further purification. In 50 μl, the ligation mixture contained 40 pmol of purified DNA-compound-DNA

conjugate, 100 pmol of phosphorylated code, 100 pmol of adapter DNA to support the aligning of target DNAs by hybridization, 1 mM ATP, and 350 units of T4 ligase. The ligation process was performed at 16 °C for 18 h. The ligation process was monitored by denaturing urea PAGE and only successfully encoded DNA-compound-DNA conjugate was purified from the gel using the protocol described above.

### DNA purification from polyacrylamide gel

The gel band containing correct-size DNA was sliced and chopped into fine particles and immersed with three-fold volumes of water in 2 ml Eppendorf tubes. The sample was frozen overnight at −20 °C and dissolved at 90 °C on a heat block for 5 min. Subsequently, the tube was stirred at 50 °C on a shaker for 18 h at 300 rpm. Next, the supernatant was collected and concentrated by extracting it against n-butanol. Repetitive extraction by removing the water-containing n-butanol layer was performed until the volume of the DNA-containing lower layer was small enough for subsequent ethanol precipitation or purification by silica membrane (QIAGEN, Netherlands). The DNA-templated reaction product was purified by ethanol precipitation and the encoded library member was purified via a silica membrane-based kit (QIAquick Nucleotide Removal Kit, QIAGEN) according to the manufacturer's instructions. 20 pmol of purified DNA-compound-DNA conjugate was injected in UPLC-ESI-MS to measure the molecular weight and confirm the purification of the correct product.

### Monitoring the assembly of three sub-libraries by native DNA polyacrylamide gel

Sub-libraries A, B, C (0.4 pmol each) were annealed in 20 μl of 1× PBS by incubating at 95 °C for 5 min and allowing to cool to RT over 2 h.

Composition of the 6% TBE gel: 1.6 ml of 5× TBE buffer, 0.96 ml of 50% stock gel solution, 40 μl 10% (m/v) ammonium persulfate, 4 μl TEMED, and water up to 8 ml.

The mixed solution was added to a mini gel cassette (Thermo Fisher Scientific, Waltham, USA) and left still for 20 min for polymerization. After polymerization, the cassette was loaded in a gel electrophoresis chamber (Bio-Rad Laboratories, Hercules, USA). 1× TBE buffer was added to the inner and outer chambers. Before loading DNA samples, the empty gel was run for 15 min in 1× TBE buffer under a constant voltage of 70 V to make the system homogeneous. Then, DNA samples were mixed with 10× loading dye in 9:1 volume, loaded on the gel, and run for 3 h. Next, the gel was detached from the cassette, stained with 1× SYBR Green II for 15 min, and imaged with ChemiDoc MP (Bio-Rad Laboratories, Hercules, USA).

### Immobilization of target proteins on a solid support for selection

2 × 3 mg (300 μl) of Dynabeads MyOne Carboxylic Acid (Invitrogen, Thermo Fisher Scientific, 65011) were washed twice with 25 mM 2-ethane sulfonic acid (MES), pH 6 in equal volume by incubating the beads in separate tubes for 10 min on a rotary mixer. EDC and NHS were dissolved in cold 25 mM MES, pH 6 at a concentration of 50 mg/ml prior to use. To the washed beads, 50 μl of EDC and 50 μl of NHS solution were added and incubated with mild tilt rotation for 30 min at RT. After incubation, the tubes were placed on the magnet holder for 4 min to remove the supernatant. The beads were then washed twice with 25 mM MES, pH 6 buffer. To one of the tubes, 100 μl of protein (3.5 mg/ml of trypsin and CAII) solution in 25 mM MES, pH 6 buffer was added to capture the protein on the beads, while 50 mM Tris, pH 7.4 was added to the other tube to generate non-coated beads. The reaction was performed for 2 h at RT with mild tilt rotation. After removing the supernatant, the beads were further incubated with 100 μl 50 mM Tris, pH 7.4 for 15 min, to block unreacted activated carboxylic groups. The beads were then washed three times with 1 ml PBS-T (0.1% Tween 20), re-suspended in 150 μl PBS-T, and stored at 4 °C.

## Affinity selections against bovine carbonic anhydrase II (CAII) and bovine trypsin

Affinity maturation of hits for CAII and trypsin was performed by mixing three sub-libraries with the 1:1:1 ratio, with each of the sub-library B at 0.05 nM in 100 µl selection volume. Non-hit and non-target control selections were performed in parallel to eliminate promiscuous binders.

20 µl of coated and uncoated beads were washed three times with 1 ml of PBS-T (1× PBS + 0.05% v/v Tween 20) and incubated with 10 µg/ml herring sperm DNA and respective library composition in 100 µl selection volume for 1 h at RT. Next, the beads were washed three times with 1 ml PBS-T to remove unbound library members. The beads were suspended in 100 µl elution buffer (10 mM Tris-Cl, pH 8.5) and the bound components were eluted from the protein by heating at 95 °C for 5 min. The eluted library members were then analyzed by quantitative real-time PCR.

## Quantitative real-time PCR (qPCR) for analysis of selection output

Primer pairs specifically amplifying each code sequence of the sub-library B as well as the primer pairs quantifying the total enrichment of sub-library A, B, and C was used to unveil the enrichment pattern of three selections for CAII and trypsin, respectively. 10 µl of reaction mixture contained 5 µl of 2× SYBR Green I master mix (Quantabio, Massachusetts, USA) 1 µl of primer pair (500 nM final concentration), 3 µl of MilliQ water, and 1 µl of the selection output. qPCR was performed with the following protocol: 10 min at 95 °C, then 40 cycles of: 15 sec at 95 °C, 1 min at 53 °C, and 30 sec at 68 °C (SYBR Green I signal acquired at this step), and final extension at 68 °C for 2 min, followed by melting procedure from 60 to 95 °C measuring decreasing fluorescence signal at a constant interval of time points. A standard curve was generated using a series of known concentrations to correlate with the acquired Ct values. Then the Ct value corresponding to each enriched compound of sub-library B was converted to an enriched amount based on the standard curve (Supplementary Tables 2, 3 and Supplementary Fig. 52).

## Purification and characterization of off-DNA small molecules

The purification of small molecular compounds was performed by reverse-phase HPLC (Waters, USA) equipped with a Luna 5 µ C18 (2) 100 Å, 100 × 10.00 mm (Phenomenex, CA, USA) using MilliQ/0.1% TFA and acetonitrile/ 0.1% TFA. The gradient varied depending on the characteristics of the reaction product and the reaction crude. The measurement of the compounds was performed by UPLC-ESI-MS (Waters, USA) equipped with an ACQUITY UPLC BEH C18 1.7 µm 2.1 ×50 mm reverse-phase column (Waters, Milford, MA, USA) as stationary phase using a linear gradient from 100% MilliQ/0.1% formic acid to 100% acetonitrile/0.1% formic acid. The For $^1$H NMR characterization, compounds were dissolved DMSO-d6 and the data were collected by Bruker AV III 600. The data are reported in terms of chemical shift (ppm), multiplicity, coupling constant, and signal integration.

## Carbonic anhydrase II inhibition assay

CAII inhibition assay was performed to measure the inhibitory effect of re-synthesized CBS-conjugates. The esterase activity of CAII with a chromogenic substrate p-nitrophenyl acetate (pNPA) was measured. The rates of hydrolysis were determined by an increase of absorption at 410 nm after incubating different compounds with CAII. Re-synthesized compounds were diluted in a serial spanning mM to µM concentrations. To 94 µl 1× PBS, pH 7.4 buffer containing 650 nM CAII in each well was added 1 µl of compounds of each concentration in a 384-well plate, and the plate was incubated for 30 min prior to the measurement. 5 µl of pNPA (50 mM in 20% DMSO) was added to the plate right before the measurement. The absorption was measured at

37 °C each 10 sec for 15 min using a Beckman Coulter's Paradigm Detection Platform (Brea, USA). The initial $V_{max}$ was obtained from the increase of absorption and the $V_{max}$ was plotted against the inhibitor concentrations and logistic fitting was performed to obtain $IC_{50}$ values using Origin 2019b (OriginLab) software.

## Trypsin inhibition assay

The inhibition by the benzamidine derivatives against trypsin was measured using an assay probing the enzyme proteolytic activity on the substrate Nα-Benzoyl-DL-arginine-4-nitroanilide hydrochloride (BApNA). Trypsin (2 µM, in PBS, pH 7.4) was incubated with different concentrations of benzamidine derivatives (via a two-fold serial dilution, 10 data points) for 30 min. 5 µl of BApNA solution (10 mM, dissolved in DMSO) were added immediately prior to the measurement. The final total volume was 100 µl per well. The measurements were performed in a Low Binding 384-well plate. The absorption was measured at 410 nm in intervals of 10 sec for 15 min at 37 °C using a Beckman Coulter's Paradigm Detection Platform (Brea, USA). The measurements were performed in triplicate. The activity of trypsin was calculated by the hydrolysis rate of BApNA after subtraction of the background-hydrolysis rate. The curves were plotted as a function of the concentration of the inhibitor against the relative enzyme activity. $V_{max}$ was derived from the detected absorption signal for each concentration and logistic fitting was performed to obtain $IC_{50}$ values using Origin 2019b (OriginLab).

## Immobilization of MMP-2/MMP-9 on solid support

4 µg of MMP-2/MMP-9 was dissolved in 100 µl of selection buffer containing 25 mM Tris-base, 10 mM CaCl₂, 150 mM NaCl, 0.05% Tween 20, pH 7.5. 10 µl of 4-Aminophenylmercuric acetate (APMA) (100 mM) solution was added to the protein solution and incubated at 37 °C for 1 h. After activation, the solution was added on top of 40 µl pre-washed Ni-NTA resin and incubated for 30 min on a rotary shaker. The supernatant was discarded, and the resin was washed three times with the selection buffer.

## Assembly of the chemically diverse trio-pharmacophore library

For each independent selection, three sub-libraries were mixed in the selection buffer with a final concentration of 40 nM of each sub-library in 100 µl. The mixture was heat denatured at 90 °C for 1 min and slowly cooled down to allow the formation of the trio-pharmacophore library. The dynamic dual-pharmacophore DEL was provided by DyNAbind GmbH (Merck, DYNA001).

## Selection against MMP-2/MMP-9

The assembled library was incubated with the protein-bound solid support for 2 h at RT on a rotary shaker. In parallel, selection on bare solid support served as the negative control to exclude promiscuous binders on the resin. After panning, the supernatant was discarded, and the resin was washed three times with 1 ml selection buffer. Finally, 100 µl of 50 mM Tris buffer, pH 7.4 was added to resuspend the resin and the resin was incubated at 95 °C for 5 min to denature the protein, release, and collect the bound library members.

The eluted library members were subjected to sample preparation for NGS. NGS preparation was achieved by two-step PCR. First, the individual selection was indexed using primers containing unique sequences corresponding to each selection. The amplified product was purified from 2% agarose gel. Second, the purified DNA from all selections were pooled in equal amount and subjected to the 2$^{nd}$ PCR step to attach sequences compatible with the NGS flow cell and the sequencing primers. NGS was performed by Novogene UK with Novaseq 6000. Raw data files (fastq files) obtained were decoded using a custom python script and excel. The sequence reads corresponding to each sample was retrieved by searching for the correct index unique to each selection. Then, the code region was extracted

from each sequence and assigned to the corresponding identity. The count of each code in each selection sample was obtained by looping through the total reads and counting the occurrence of each code. The enrichment fold of each compound was calculated by first, dividing the count by the total count to get the abundance, second, the abundance after the selection was divided by the abundance of pre-selection to reflect the distribution change of the compound. Next, the enrichment fold was plotted against the count for each member of the library to give the scatter plots as in Fig. 4.

## MMP-2 and MMP-9 inhibition assay

The inhibition by the small molecule compounds against MMP-2/MMP-9 was measured using an assay probing the enzyme proteolytic activity on the substrate DNP-Pro-Leu-Gly-Met-Trp-Ser-Arg (Enzo Life Sciences, USA). All assays were performed in the assay buffer containing 50 mM Tris, 100 mM NaCl. 5 mM $CaCl_2$, and 0.1% Brij 35, pH 7.5. Human proMMP-2 and proMMP-9 were purchased from Sino biological (Germany). The proMMPs were activated by 1 mM APMA in the assay buffer at 37 °C for 2 h. After activation, MMP-2 was diluted to 6.5 nM of final assay concentration, and MMP-9 was diluted by the buffer to reach 10 nM of final assay concentration. In each well, 49 μl of enzyme solution was incubated with 1 μl of compounds of series of concentration (nM-mM in DMSO) at RT for 30 min. 2 μl of the FRET substrate was added right before the measurement at a final concentration of 25 μM. The rate of hydrolysis was monitored by quantifying the emission at 360 nm (ex: 280 nm). The emission was measured with intervals of the 30 s for 30 min with Synergy H1 Plate Reader (Agilent, USA). The measurements were done in triplicate. At each concentration of compound, the rate of hydrolysis at the initial stage ($V_{max}$) was calculated and relative enzyme activity was obtained by normalizing the data without inhibitor to 1. Then, curves were plotted as a function of the concentration of the inhibitor against the relative enzyme activity. $V_{max}$ was derived from the detected absorption signal for each concentration and logistic fitting was performed to obtain $IC_{50}$ values using Origin 2019b (OriginLab).

## On-DNA hit validation of ligand pairs binding to alpha-1-acid glycoprotein via biolayer interferometry (BLI)

5 L ref DNA: Amino-C6-GGAGGTGTAGACGACAGAGTATTTG

 3 L ref DNA: CTCGATCTGGCTGCGATCCCAACCTCC-C6-Amino

 P2 anchor DNA: Amino-C6-GAGATCGGAAGAGCGTCG

 BLI adapter DNA: CGACGCTCTTCCGATCTCCAAATACTCTGTCG TCTACTGGGATCGCAGCCAGATCGAG

 Functionalization Buffer: 100 mM imidazole, pH 6

 Loading Buffer: 10 mM HEPES, pH 7.2, 150 mM NaCl, 0.05% Tween® 20

 Regeneration Buffer: 10 mM HEPES, pH 7.21% SDS

 PBS-T: 10 mM $Na_2HPO_4$, 1.8 mM $KH_2PO_4$, pH 7.4, 2.7 mM KCl, 137 mM NaCl, 0.05% Tween® 20

 The respective 5L-DNA-conjugate, 3L-DNA conjugate, and BLI adapter DNA were mixed in Loading Buffer at a final concentration of 1 μM each. Then, the DNA–fragment pair construct was annealed by heating to 60 °C for 5 min, followed by a slow cool down to room temperature.

 The BLI experiments were carried out using the Octet® RED384 interferometer system with Amine Reactive 2nd Generation (AR2G) Dip and Read™ Biosensors. After initial hydration in water for 10 min, the biosensors were functionalized by activating with 200 μM EDC (in Functionalization Buffer) for 300 s and loading with 100 μM P2 anchor DNA (in Functionalization Buffer) for 600 s. The activation–loading cycle was repeated twice. After the third activation–loading cycle, the sensors were quenched in 1 M ethanolamine for 300 s. The functionalized sensors were loaded with 100 nM DNA–fragment pair construct (in Loading Buffer) for 600 s. The sensors were then dipped into Regeneration Buffer for 20 s. For

the kinetic measurement, the sensors were equilibrated in PBS-T for 300 s, followed by a baseline measurement in fresh PBS-T for 60 s. To measure binding association, the sensors were then dipped into a 2 μM AGP solution (in PBS-T) for 100 s. The dissociation was measured by dipping the sensors into the PBS-T solution previously used for baseline establishment for 300 s. Finally, the sensors were regenerated by 3× alternatingly dipping in Regeneration Buffer and PBS-T for a total of 30 s. The kinetic measurement (from equilibration to regeneration) was then repeated another two times at higher AGP concentrations (10 and 50 μM) using the regenerated sensors. A "blank" DNA construct not containing a fragment pair was used as a reference. The BLI data was analyzed and exported using the FortéBio Data Analysis 9.0 software.

## Molecular docking studies

The crystal structures of CAII (PDB: "6SKV"), bovine trypsin (PDB: "1BTY"), MMP-2 (PDB: "1QIB"), MMP-9 (PDB: "4H3X"), and AGP (PDB: "3KQ0") were downloaded from the Protein Data Bank.

 Solvent molecules, duplicated chains, and bound ligands were removed from the crystal structures. Hydrogens were added to the receptors, and then the PDB files were prepared into pdbqt files with the ADFR software suite. Ligands were prepared to pdbqt files with the Meeko python package. Grid boxes around the binding site of the receptors were defined according to the known ligands in the receptor structures. The grid centers were defined as the centers of the known ligands, and the box size was defined as $60 \times 60 \times 60$ grid points ($22.5 \times 22.5 \times 22.5$ Å). Then the affinity maps were created using AutoGrid4. CAII, MMP-2, and MMP-9 are zinc metalloenzymes in which zinc ion plays an important role. Thus, a specialized force field, the AutoDock4 Zn force field[56], was applied to the zinc ions of these zinc metalloenzymes by adding tetrahedral zinc pseudo atoms. The docking process was performed using AutoDock Vina[4] with AD4 scoring, and the exhaustiveness was adjusted between 8 and 32. Finally, the best binding poses were selected for ligand-receptor interaction analysis using the Hbind package[57]. Distances between the interested groups of the compounds and the surrounding residues were calculated using the pytraj python package. OpenBabel was used to convert molecule formats, and PyMOL was used to create the ligand-receptor interaction views.

## Statistics and reproducibility

All experiments were performed in three biological repeats. Data are shown as mean and SD or SE. Statistical analyses were performed with Origin 2019b (OriginLab). No statistical method was used to predetermine the sample size. The experiments were not randomized.

## Reporting summary

Further information on research design is available in the Nature Portfolio Reporting Summary linked to this article.

## Data availability

The supporting information is available as a separate file. Raw numerical data underlying all graphs, as well as raw mass spectrometry data, are supplied as source data. Next-Generation Sequencing raw data is available at the SRA database with the accession number "PRJNA887468". Source data are provided with this paper.

## Code availability

The custom python scripts for decoding NGS results are provided in Supplementary Note 4.

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

## Acknowledgements

We thank Ulrike Hofmann for technical support and Dr. Kathrin Putke from DKMS Life Science Lab (Germany), Dr. Elena Ferrari, and Meng Chen from Novogene (UK) for supporting on Next-Generation Sequencing. We thank Dr. Katerina Vafia (DyNAbind GmbH, Germany) for the support of the NGS data analysis. We thank Dr. Ekta Kumari for the technical support on the enzyme inhibition assays. This work was supported by the German Federal Ministry of Research and Education (BMBF) [03Z2EN12 and 03Z2E511]. Prof. Yixin Zhang is the recipient of the funding.

## Author contributions

M.C. and Y.Z. designed the experiments. M.C., M.P.G., and W.L. conducted the experiments. Q.C. performed all molecular docking studies. Q.C. and D.N. analyzed the docking results. D.N. performed the on-DNA BLI experiments. F.R., M.T., and S.H. synthesized the dual-pharmacophore DEL. All authors contributed to writing the manuscript.

## Funding

## Competing interests

DyNAbind GmbH is commercializing the dynamic dual-pharmacophore DNA-Encoded Library technology reported in this publication. M. Thompson, F.V. Reddavide, and Y. Zhang are shareholders of DyNAbind GmbH. The remaining authors declare no competing interests.
