## [Peer Review File · Nature Communications]

REVIEWER COMMENTS

Reviewer #1 (Remarks to the Author):

The manuscript of Reddavid, Chen, Zhang and co-workers describes the construction of a DNA-encoded chemical library (DEL) which is based on both a DNA-templated approach (as described first by Liu and coworkers: Gartner, Z. J. et al. (2004) *Science* 305, 1601–1605)

and on a self-assembling strategy ("ESAC", first described by Neri and coworkers in 2004, later refined by Neri, Scheuermann and coworkers (Wichert, M. et al. (2015) *Nat Chem* 7, 241–249)

(2015) and, in a dynamic fashion, by some of the authors themselves (Reddavid et al. (2015) *Angew Chem Int Ed Engl.* 54(27), 7924-8).

The concept relies on the self-assembly of three sub-libraries, two of which (DEL-A and DEL-C) containing one diversity element and resembling the sublibraries used for ESAC), while the third sublibrary DEL-B is constructed from bifunctional fragments by DNA-templated synthesis using a set of DEL-compatible reactions which are fused together on two partially complementary oligonucleotides, serving as "linker" moieties.

Eventually, DEL-B assembles with both DEL-A and DEL-C and forms a "tri-pharmacophore" assembled DEL, whereby the pharmacophores are not covalently connected. The authors clearly show the formation of the desired trio-DEL on gel and by MS and constructed a library with a size of 23.5 Mio members (883x30x890).

The trio-DEL was then tested for performance on carbonic anhydrase II and trypsin as well-established targets and further used for de novo selections on the two matrix metalloproteinases MMP2 and MMP9.

For the test targets CAII and trypsin known week-binding fragments were included in the "external" sublibraries DEL-A and DEL-C and the right fragment combination was indeed preferentially selected, in combination with various "internal" DEL-B linker moieties. While these linker moieties led to an improvement over the better of the two known fragments by roughly one order of magnitude with various DEL-B linkers, it would be interesting to see the direct comparison with a standard dual-pharmacophore DEL containing the two fragments.

In the presented Trio-setup, the inner DEL-B linker library serves for spacing the two externally displayed fragments more far compared with the dual-pharmacophore approach, which might be good or bad, with the additional advantage that the inner linking fragment may further lead to additional binding interactions with the protein target (as can also be seen from the presented docking studies). For CAII the authors used the fragment 3-{5-[3-(trifluoromethyl) phenyl]-2-furyl} acrylic acid which they had previously identified using dynamic self-assembling library technology. The same fragment was also reported as a fragment binding to alpha1-glykoprotein (AGP) by Wichert et al. (see above) and by the authors themselves. It would be interesting to see the performance of the new "trio" DEL on AGP, as this is probably the best control for dual-pharmacophore selections so far.

The new ligands reported for the very difficult "de novo" targets MMP2 and MMP9 with inhibition constants in the double-digit micromolar range indeed suggest that this method may prove useful, as it increases the overall interaction interface with the target compared with the neat dual-pharmacophore setup featuring two small fragments only. It is interesting to see that for both MMPs the same fragments were identified from DEL-A and DEL-C, and also the same linkers from DEL-B.

As the authors noted a proper encoding technology for the tri-pharmacophore system, in analogy to what had been proposed by Wichert et al. for the dual-pharmacophore approach is still missing, which hampers the identification of simultaneously selected fragments and linkers. Without such encoding scheme each sublibrary needs to be decoded individually and the relevant combinations of enriched compounds have all to be tested.

While the new approach indeed might shed new light on a potential linking with "inner fragments" the compounds which are being selected are not covalently linked, meaning that also in this system the "actual linker" leading to a covalent hit still needs to be explored post selection.

Minor points:

- the reference for the first DNA-encoded self-assembling chemical library is missing (Melkko S, Scheuermann J, Dumelin CE, Neri D. (2004) Encoded self-assembling chemical libraries. *Nat Biotechnol.* 22(5):568-74) is missing and should be given in line 48 of the Introduction
- Introduction, line 62: The potential triplex DEL had also already been proposed by the same publication
- The results of the dual-pharmacophore selection against the MMPs (Figure S3D) should be discussed in comparison to the trio-pharmacophore selection.
- The structures of the building blocks and codes of the sub-libraries should be given as SI

In summary, the proposed trio-pharmacophore setup is a very interesting expansion of the DNA-encoded self-assembling chemical library field and it yielded promising results. In my opinion, it deserves being published by *Nat Commun*, after addressing the points outlined above. The manuscript itself as well as the SI are well written.

Reviewer #2 (Remarks to the Author):

In this manuscript, the authors describe the application of a trio-pharmacophore DEL (T-DEL) for the discovery and optimization of fragments and linkers in fragment-based drug discovery (FBDD). The authors detail the design of a sub-library of linkers/fragments flanked by two DNA strands used to

assemble the other two fragment library components for the T-DEL. While the concept of a triplex DEL was previously postulated, this is the first report of a T-DEL of over 23 million members that could be implemented as a de novo hit finding platform, therefore, it is of practical value and interest to the DEL field and FBDD field.

Key strengths and noteworthy results

The paper describes a useful adaptation of DNA-templated synthesis of the linker/fragment sub-library that enables the design, assembly and purification of a multi-million member T-DEL adequate for selection experiments. Affinity maturation selection was designed and performed using known fragments with the linker sub-library for method validation. Post selection synthesis of the putative binders resulting from fragment combination and activity assays were carried out for hit confirmation. Molecular docking studies were performed to explain the possible binding interactions.

Potential issues and further experiments

1. The authors describe their approach of utilizing T-DEL for the linker optimization of known fragment pairs. However, in the event, the entire SL-B (the linker sub-library) was enriched. This indicates fragment(s) facilitated interaction with little information indicating linker preference, nor any correlation of the trend of enrichment to the assay activity of the linked fragment compounds (C0-C5). Although docking studies were undertaken in order to illustrate the binding interactions of linker fragments C0-C5, relevant plausibility needs to be added to illustrate how linker optimization was informed by the observations in both CAII and bovine trypsin positive control T-DEL selection experiments.
2. In the section of de novo selections, a dynamic dual-pharmacophore DEL with the same members of SL-A and SL-C was performed against MMP-2 and MMP-9. Please explain the purpose of this experiment.
3. There is a C6 or C7 linker between fragments and oligonucleotides in the library construct, which allows individual flexibility of the fragments leading to a large number of possible fragment interactions in the binding pockets. Because the fragment (A)-linker/fragment (B)-fragment (C) in the T-DEL is not covalently linked, an A-B-C array of 45 resynthesized off-DNA compounds overly simplifies the multiplicity of actual binding interactions of these compounds. The geometry of fragment, linkage points, and the order and the direction of how A-B-C are linked should be considered in the design of the A-B-C array. In addition, the enriched fragments of three sub-libraries were individually decoded, the array design should also include the combination of AB, BC, AC directionally. One may hypothesize that the binding pose of B can be informed by the enrichment of SL-B. This has to be experimentally confirmed with the encoded reversed non-symmetric Bs with flanked DNA. Regardless, the hit confirmation of a trio-pharmacophore approach is complicated and highly labor intensive. The authors will have to justify why the approach is advantageous over other methods in FBDD.

4. Enrichment in selection is indicative of a binding event of fragments to a target of interest. It is worth pointing out the disconnect when fragment hits are evaluated in activity assays.

Reviewer #3 (Remarks to the Author):

The manuscript of Cui et al. describes a method for producing DNA-encoded chemical libraries containing tri-pharmacophores. More accurately, the method combines DNA-encoding with fragment-based drug discovery. The approach is very clever and allows for purification of sub-libraries via PAGE. The purification step ensures that synthetic efficiency is high, but can also be viewed as a somewhat cumbersome step in the process. The identification of "hits" is followed by optimization of linkages between fragments.

The libraries have been used to identify potential inhibitors for carbonic anhydrase II, trypsin, MMP-2, and MMP-9. In the case of carbonic anhydrase II, an inhibitor with an IC₅₀ value of 670 nM. This is okay for a starting point. For MMP-2 and MMP-9 inhibitors were identified with IC₅₀ values in the 10-15 micromolar range. This is not particularly compelling.

The overall focus of this study is the method to create the tri-pharmacophore libraries, which is well-executed. Nonetheless, the results from using this approach did not yield initial inhibitory compounds that showed distinct advantages over prior approaches.

We want to thank the reviewers for their insightful comments and suggestions. During revision, we investigated the performance of T-DEL by benchmarking the dual pharmacophore library using three model targets: carbonic anhydrase II (CAII), trypsin, and alpha-1-acid glycoprotein (AGP), and their well-characterized ligand pairs (Figures R1-R5). We have included the results and corrections in the main text and supporting information of the revised manuscript. We changed the authorship of Dr. Dzung Nguyen to the second author due to his performance of ligand binding assays. Dr. Qinchang Chen has started a new position at the Research Institute of Intelligent Computing, Zhejiang Lab, during revision; therefore, we added the new affiliation.

The following are the point-by-point responses to each comment (the reviewer's comments are in blue, our responses are in black, and the changes in the main text are highlighted in red):

Reviewer 1

The manuscript of Reddavid, Chen, Zhang and co-workers describes the construction of a DNA-encoded chemical library (DEL) which is based on both a DNA-templated approach (as described first by Liu and coworkers: Gartner, Z. J. et al. (2004) *Science* 305, 1601–1605) and on a self-assembling strategy ("ESAC", first described by Neri and coworkers in 2004, later refined by Neri, Scheuermann and coworkers (Wichert, M. et al. (2015) *Nat Chem* 7, 241–249) (2015) and, in a dynamic fashion, by some of the authors themselves (Reddavid et al. (2015) *Angew Chem Int Ed Engl.* 54(27), 7924-8).

The concept relies on the self-assembly of three sub-libraries, two of which (DEL-A and DEL-C) containing one diversity element and resembling the sublibraries used for ESAC), while the third sublibrary DEL-B is constructed from bifunctional fragments by DNA-templated synthesis using a set of DEL-compatible reactions which are fused together on two partially complementary oligonucleotides, serving as "linker" moieties.

Eventually, DEL-B assembles with both DEL-A and DEL-C and forms a "tri-pharmacophore" assembled DEL, whereby the pharmacophores are not covalently connected. The authors clearly show the formation of the desired trio-DEL on gel and by MS and constructed a library with a size of 23.5 Mio members (883x30x890).

The trio-DEL was then tested for performance on carbonic anhydrase II and trypsin as well-established targets and further used for de novo selections on the two matrix metalloproteinases MMP2 and MMP9.

For the test targets CAII and trypsin known weak-binding fragments were included in the "external" sublibraries DEL-A and DEL-C and the right fragment combination was indeed preferentially selected, in combination with various "internal" DEL-B linker moieties. While

these linker moieties led to an improvement over the better of the two known fragments by roughly one order of magnitude with various DEL-B linkers, it would be interesting to see the pharmacophore DEL containing the two fragments.

We fully agree with the reviewer's comment. We investigated and compared the enrichments of pairs of known binders in dual- and trio-pharmacophore formats using two model proteins (CAII and trypsin). As shown in Figure R1, we measured the recovery of the ligand pair from selections against CAII and trypsin using qPCR. Each sub-library was added at the final concentration of 1 nM per selection in 100 μ l volume. The recovery of each sub-library was measured by qPCR. The assays were performed in dual- and trio-pharmacophore formats in parallel (Figure R1A), and each selection was performed in triplicates. As observed in Figures R1C and D, the ligand pair of CAII (CBS and compound A) displayed higher recovery than that of trypsin in target selection, agreeing with the measured affinity (Figures 2C and 3C). Both dual- and trio-pharmacophore formats delivered comparable recovery of the ligand pair of CAII (Figure R1C). However, the lower-affinity ligand pair 4-aminomethyl benzamidine and compound B enriched less in trio-pharmacophore format (Figure R1D). A similar pattern was also observed in another target alpha-1-acid glycoprotein (Figure R4).

In this comparison between one ligand pair (in dual-pharmacophore format) and the mixture of 30 different combinations (in T-DEL format), we can conclude that most members of the linker library (SL-B) do not improve, even impair the binding. The signals from T-DEL represent the averages of 30 different combinations, and due to the large difference among different SL-B members in enrichment, only a few members from SL-B can improve the binding. Notably, in our T-DEL design, library A cannot assemble with library C without library B (Figure 1E).

Figure R1 (Corresponding to Figure S8). Comparison of T-DEL format of 30 combinations to the dual-pharmacophore format of one pair of ligands with model targets. (A) Schematic illustration of libraries in dual- and trio-pharmacophore formats. The ligand pairs were displayed as single-member sub-libraries, respectively. In T-DEL format, the two sub-libraries A and C were assembled with the 30-member sub-library B. (B) The ligand pairs of CAII and trypsin are shown in the table. (C) Recovery of each sub-library from the affinity selection against CAII (left) and blank solid support (right). (D) Recovery of each sub-library from the

affinity selection against trypsin (left) and blank solid support (right). SL-A, SL-B, and SL-C are from T-DEL format, and 5L and 3L are from dual-pharmacophore format.

In the presented Trio-setup, the inner DEL-B linker library serves for spacing the two externally displayed fragments more far compared with the dual-pharmacophore approach, which might be good or bad, with the additional advantage that the inner linking fragment may further lead to additional binding interactions with the protein target (as can also be seen from the presented docking studies).

Thank you for this insightful comment. You are absolutely right that the sub-library B members may or may not bring an additional constructive interaction, and the measured total recoveries in Figures R1 and R4 represent the average of 30 combinations, especially when weak fragment pairs are displayed externally, e.g., the ligand pairs of trypsin (Figure R1D) and AGP (Figures R2 and R4), the trio-pharmacophore format displayed overall lower recovery compared to the dual-pharmacophore format. Nevertheless, the enriched individual sub-library B members proved advantageous by the off-DNA resynthesis and the enzyme inhibition assay (Figures 2 and 3).

For CAII the authors used the fragment 3-{5-[3-(trifluoromethyl) phenyl]-2-furyl} acrylic acid which they had previously identified using dynamic self-assembling library technology. The same fragment was also reported as a fragment binding to alpha1-glycoprotein (AGP) by Wichert et al. (see above) and by the authors themselves. It would be interesting to see the performance of the new "trio" DEL on AGP, as this is probably the best control for dual-pharmacophore selections so far.

We agree that AGP and its reported ligand pair are indeed a good model to evaluate the performance of T-DEL. As shown in Figure R2B, we applied two reported fragment pairs: Biphenyl-2-yl-(toluene-4-sulfonyl)-amino)-acetic acid (Compound C) paired with 3-[5-(2-(Trifluoromethyl)phenyl)furan-2-yl]-acrylic acid (Compound A) and Compound C paired with 3-[5-(2-(Trifluoromethyl)phenyl)furan-propanoic acid (Compound D) (Bigatti, M. et al. (2017) ChemMedChem 12(21), 1748-1752, and Wichert, M. et al. (2015) Nat. Chem. 7, 241–249) whose affinity differed by one order of magnitude in an on-DNA binding assay by Octet biolayer interferometry (BLI) (Figure R2).

Figure R2. (Corresponding to Figure S9) Alpha-1-acid-glycoprotein (AGP) as a model for evaluating DELs in different formats. (A) The reported two ligand pairs binding to AGP. (B) On-DNA affinity measurements by Octet Biolayer Interferometry (BLI). The two compounds of each ligand pair were displayed at the 5' and 3' termini of a dsDNA, respectively. Each construct was loaded on the BLI sensor, and the binding of AGP to the sensor surface was measured, which led to the determination of K_{on} and K_{off} values.

Molecular docking studies revealed compound C binding to the same site as the reported ligand (PDB: 3KQ0). When the two compounds of the ligand pairs were docked together to AGP, A or D occupied a sub-site adjacent to that of C. Both ligand pairs displayed the same binding mode, while C+D exhibited a better docking score, presumably due to the rotatable single bond of D compared to A (Figures R3 and R5).

Figure R3. (Corresponding to Figure S10) Simultaneous docking of the ligand pairs (C+A and C+D) in complex with AGP (PDB ID: 3KQ0). Yellow dashed lines indicate hydrogen bonds.

With the two ligand pairs binding to the same target with different affinities, we again measured the recoveries from the selections in dual- and trio- pharmacophore formats (Figure R4). As shown in Figures R2 and R4, compounds C and D exhibited approx. 40 times higher affinity than the other pair in the binding assay and recovered higher in the selection in both formats. The dual pharmacophore format led to a higher recovery of the ligand pair compounds C and A, while both formats showed comparable enrichment with the high-affinity pair compounds C and D. Interestingly, we have observed that with the increase of binding affinity of ligands in SL-A and SL-C, the overall contributions from SL-B on binding can be augmented, as shown by the ligand-dependent enhancement of recovery in the T-DEL format.

As discussed above, the signals from T-DEL selections are the averages of 30 different combinations, while the concentration of each is 0.033 nM, in contrast to the 1 nM of the dual pharmacophore format. Therefore, we further measured the enrichment of each sub-library B member and found that some members were selectively enriched in target selection but not in blank control selection. Further, in the T-DEL selections with either C+A or C+D, we observed overlapping hits from SL-B (16, 19, and 20) (Figures R4C and R4D).

Figure R4. (Corresponding to Figure S11) Comparison of T-DEL format to the dual-pharmacophore format with alpha-1-acid glycoprotein (AGP). (A) The ligand pair Compound C and Compound A served as SL-A/5L and SL-C/3L, respectively. SL-A and SL-C were assembled with 30-member SL-B to form the T-DEL. (B) The ligand pair Compound C and Compound D served as SL-A/5L and SL-C/3L, respectively. SL-A and SL-C were assembled

with 30-member SL-B to form the T-DEL. (C and D) The abundance of each member in SL-B in naïve pre-selection library, no-target selection, and target selection (left). The boxes range from 25 to 75 percentile. Members of pronouncing abundance or enrichment are labeled with their IDs. The horizontal lines indicate the median. Enrichment profile of each member in SL-B by normalizing to naïve library and no-target selection (right). Abundance was calculated by dividing the recovered amount of each member by the total recovered amount of SL-B. The Enrichment Fold was calculated in two steps. First, the amount of total SL-B from target selection was divided by the amount of total SL-B from no-target selection to calculate the enrichment factor (F). Second, the abundance of each member from target selection was divided by the abundance of each member from no-target selection, and the obtained value was multiplied by F to deliver the Enrichment Fold.

With the selected hits 16, 19, and 20, we virtually synthesized six linked compounds (C-16-A, C-19-A, C-20-A, C-16-D, C-19-D, and C-20-D) and performed molecular docking (Figure R5). All compounds were binding to the same binding site as C+A and C+D. In particular, C-16-A and C-16-D displayed better docking scores and larger hydrophobic contacts than compounds linked by 19 and 20 (Figure R5).

Figure R5. (Corresponding to Figure S12) **(A)** Summary of interactions between small molecule compounds and Alpha-1-acid glycoprotein (AGP). (PDB: 3KQ0) **(B)** Structures of

the virtually synthesized compounds linked by SL-B members 16, 19, and 20. **(C)** Docking poses of the compounds in complex with AGP. Yellow dashed lines indicate hydrogen bonds.

The new ligands reported for the very difficult "de novo" targets MMP2 and MMP9 with inhibition constants in the double-digit micromolar range indeed suggest that this method may prove useful, as it increases the overall interaction interface with the target compared with the neat dual-pharmacophore setup featuring two small fragments only. It is interesting to see that for both MMPs the same fragments were identified from DEL-A and DEL-C, and also the same linkers from DEL-B.

As the authors noted a proper encoding technology for the tri-pharmacophore system, in analogy to what had been proposed by Wichert et al. for the dual-pharmacophore approach is still missing, which hampers the identification of simultaneously selected fragments and linkers. Without such encoding scheme each sublibrary needs to be decoded individually and the relevant combinations of enriched compounds have all to be tested.

A code-joining strategy is indeed needed to facilitate the identification of favorable building block combinations. Our current work is a proof-of-principle to articulate the idea of three self-assembling sub-libraries. The T-DEL is geometrically more complex than the dual pharmacophore DELs. The T-DEL structure shall be fine-tuned, and various code-joining mechanisms shall be tested to identify a construct with high code-joining efficacy and low steric interference on the ligand-protein interaction. We will work on incorporating the code-joining mechanism in a trio-pharmacophore library in due course.

While the new approach indeed might shed new light on a potential linking with "inner fragments" the compounds which are being selected are not covalently linked, meaning that also in this system the "actual linker" leading to a covalent hit still needs to be explored post selection.

We agree with the reviewer that sub-library B lacks covalent tethering and cannot represent a full linker. The resulting off-DNA resynthesized compounds need further medicinal chemistry optimization to develop into real hits. In our view, sub-library B allows us to obtain additional information about the constructive binding moiety to bridge the two flanking fragments. T-DEL remains a fragment-based drug discovery approach (FBDD), thus retaining the advantages and disadvantages of FBDD.

Minor points:

- the reference for the first DNA-encoded self-assembling chemical library is missing (Melkko S, Scheuermann J, Dumelin CE, Neri D. (2004) Encoded self-assembling chemical libraries. Nat Biotechnol. 22(5):568-74) is missing and should be given in line 48 of the Introduction.

- Introduction, line 62: The potential triplex DEL had also already been proposed by the same publication.

Thank you very much for pointing out the mistakes. We have included the missing reference in the revised manuscript (Reference 30).

- The results of the dual-pharmacophore selection against the MMPs (Figure S3D) should be discussed in comparison to the trio-pharmacophore selection.

We fully agree with the reviewer. The distinctive hits were enriched in common from both formats, which made us more confident in selecting them for off-DNA synthesis. Therefore, we added the sentence, "Remarkably, we identified common hits using both formats of the same building blocks, indicating that these fragments can be specifically enriched independent from the library design, making it more confident for us to consider them as true positive hits." in the revised manuscript.

- The structures of the building blocks and codes of the sub-libraries should be given as SI.

The dual-pharmacophore library was provided by the company DyNABind GmbH, and the full list of building blocks of SL-A and SL-C cannot be provided. The full list of SL-B is given in Figure S2.

In summary, the proposed trio-pharmacophore setup is a very interesting expansion of the DNA-encoded self-assembling chemical library field and it yielded promising results. In my opinion, it deserves being published by Nat Commun, after addressing the points outlined above. The manuscript itself as well as the SI are well written.

Reviewer #2 (Remarks to the Author):

In this manuscript, the authors describe the application of a trio-pharmacophore DEL (T-DEL) for the discovery and optimization of fragments and linkers in fragment-based drug discovery (FBDD). The authors detail the design of a sub-library of linkers/fragments flanked by two DNA strands used to assemble the other two fragment library components for the T-DEL. While the concept of a triplex DEL was previously postulated, this is the first report of a T-DEL of over 23 million members that could be implemented as a de novo hit finding platform, therefore, it is of practical value and interest to the DEL field and FBDD field.

Key strengths and noteworthy results

The paper describes a useful adaptation of DNA-templated synthesis of the linker/fragment sub-library that enables the design, assembly and purification of a multi-million member T-DEL adequate for selection experiments. Affinity maturation selection was designed and performed using known fragments with the linker sub-library for method validation. Post selection synthesis of the putative binders resulting from fragment combination and activity assays were carried out for hit confirmation. Molecular docking studies were performed to explain the possible binding interactions.

Potential issues and further experiments

1. The authors describe their approach of utilizing T-DEL for the linker optimization of known fragment pairs. However, in the event, the entire SL-B (the linker sub-library) was enriched. This indicates fragment(s) facilitated interaction with little information indicating linker preference, nor any correlation of the trend of enrichment to the assay activity of the linked fragment compounds (C0-C5). Although docking studies were undertaken in order to illustrate the binding interactions of linker fragments C0-C5, relevant plausibility needs to be added to illustrate how linker optimization was informed by the observations in both CAII and bovine trypsin positive control T-DEL selection experiments.

As the reviewer commented, the amount of the entire SL-B recovered from selection was higher than that of the no-ligand target selection and the no-target selection, indicating that the ligand pair and SL-B anchored specifically to the target facilitating the interaction between

SL-B and the target. All thirty members were detected after selection, which is common for DELs of small libraries, especially when the input is far larger than 10^5 copies (Sannino, A. et al. (2019), *ChemBioChem*, 20(7), 955-962; Chen, Q. et al. (2020), *SLAS Discovery*, 25(5): 523-529; Reddavid, F. et al. (2019) *Chem. Comm.*, 55(26), 3753-3756). We utilized over 10^9 copies of each member as input for all selections. Moreover, all members of SL-A and SL-C were detected in the real selections against MMP-2 and MMP-9 and showed variable enrichment fold after being normalized to the control selections, which enabled us to separate potent building blocks from others, as demonstrated by Figures 2A, 3A, 4A, and S11.

In Figure 2, the SL-B compound with the highest enrichment (C-2) displayed a 20-fold improvement in the IC_{50} value ($0.67 \mu\text{M}$). Compounds with moderate and low enrichment, C-3, C-4, and C-5, exhibited lower inhibitory effects than C1 and C2, agreeing with the selection outcome. In Figure 3, the enriched SL-B members t1, t2, t3, and t4 led to a remarkable enhancement of the inhibitory effect, especially T-2 and T-3, displaying 70-fold and 30-fold improvement, respectively.

2. In the section of de novo selections, a dynamic dual-pharmacophore DEL with the same members of SL-A and SL-C was performed against MMP-2 and MMP-9. Please explain the purpose of this experiment.

The building blocks in 5L of the dynamic dual-pharmacophore DEL are the same as the sub-library A of the T-DEL. The same holds for the 3L and sub-library C. The purpose of the experiments was to assess the ability of the new T-DEL format to identify hits in comparison to the well-established dual-pharmacophore format. Remarkably, common hits were identified with both formats against MMP-2 and MMP-9, indicating that these fragments can be specifically enriched independent from the library design, which gave us higher confidence in selecting the hits and elaborating them by off-DNA combinatorial syntheses. We have added the explanation in the revised manuscript.

3. There is a C6 or C7 linker between fragments and oligonucleotides in the library construct, which allows individual flexibility of the fragments leading to a large number of possible fragment interactions in the binding pockets. Because the fragment (A)-linker/fragment (B)-fragment (C) in the T-DEL is not covalently linked, an A-B-C array of 45 resynthesized off-DNA compounds overly simplifies the multiplicity of actual binding interactions of these compounds. The geometry of fragment, linkage points, and the order and the direction of how A-B-C are linked should be considered in the design of the A-B-C array. In addition, the enriched fragments of three sub-libraries were individually decoded, the array design should also include the combination of AB, BC, AC directionally. One may hypothesize that the

binding pose of B can be informed by the enrichment of SL-B. This has to be experimentally confirmed with the encoded reversed non-symmetric Bs with flanked DNA. Regardless, the hit confirmation of a trio-pharmacophore approach is complicated and highly labor intensive. The authors will have to justify why the approach is advantageous over other methods in FBDD.

We agree with the reviewer that the three building blocks are not covalently linked, which brings large flexibility at the binding interface. Therefore, the 45 resynthesized compounds represent a fraction of all possible binding modes. In the off-DNA syntheses, we tried our best to keep the geometry and direction of the building block in the same way as in the library. For example, the same functional groups used for on-DNA conjugations were used for the reactions with the intended building blocks. What made it more labor-intensive is the lack of a code-joining mechanism, as the reviewer pointed out. A code-joining strategy is needed to facilitate the identification of favorable building block combinations. Our current work represents a proof-of-principle to demonstrate the idea of three self-assembling sub-libraries. We will work on incorporating the code-joining mechanism in a trio-pharmacophore library in due course.

In our view, sub-library B allows us to obtain additional information about the constructive binding moiety to bridge the two flanking fragments and can guide the design of the full linkers by structure-based approaches, which is not feasible by dual-pharmacophore libraries. Compared to other methods of FBDD, the T-DEL allows us to identify three fragments simultaneously with a large combinatorial library in a simple selection process.

4. Enrichment in selection is indicative of a binding event of fragments to a target of interest. It is worth pointing out the disconnect when fragment hits are evaluated in activity assays.

Thank you for pointing out the disconnect between affinity selection and activity assays. As observed in our work, compounds 66 and 826 displayed high enrichment from the selections but did not exhibit meaningful activity in the enzyme inhibition assays. Moreover, conjugation of 66/826 to other potent fragments even diminished their activities, which may suggest that 66/826 are strong binders without exerting inhibition, and it needs validation by other biophysical tools to measure the affinity. We added the discussion in the revised manuscript.

Reviewer #3 (Remarks to the Author):

The manuscript of Cui et al. describes a method for producing DNA-encoded chemical libraries containing tri-pharmacophores. More accurately, the method combines DNA-encoding with fragment-based drug discovery. The approach is very clever and allows for purification of sub-libraries via PAGE. The purification step ensures that synthetic efficiency is high, but can also be viewed as a somewhat cumbersome step in the process. The identification of "hits" is followed by optimization of linkages between fragments.

The libraries have been used to identify potential inhibitors for carbonic anhydrase II, trypsin, MMP-2, and MMP-9. In the case of carbonic anhydrase II, an inhibitor with an IC50 value of 670 nM. This is okay for a starting point. For MMP-2 and MMP-9 inhibitors were identified with IC50 values in the 10-15 micromolar range. This is not particularly compelling.

The high-affinity small molecule MMP inhibitors reported so far typically harbor strong zinc-chelating groups such as hydroxamate and sulfonamide groups (Vandenbroucke, R. and Libert C. (2014) Nat. Rev. Drug Discov., 13(12), 904-927), which can lead to off-target inhibition of other metalloenzymes. For example, strong zinc-binding compounds such as Batimastat and Prinomastat exhibited severe toxicities in clinical trials (Mondal, S. et al. (2020) Eur. J. Med. Chem., 194, 162260). MMPs have been challenging targets, and no selective inhibitor has passed clinical trials.

As a fragment-based drug discovery approach, the T-DEL allows us to discover new binder structures by identifying three fragments simultaneously with a large combinatorial library in a simple selection process. As an early hit discovery tool, our T-DEL format successfully identified potent fragments without strong zinc-chelating groups. We agree with the reviewer that the inhibitory effect of the resynthesized compounds is not compelling and needs further optimization. We have added the sentence in the discussion section of the manuscript. In the meantime, we anticipate that by expanding the chemical space of SL-B, we may identify more potent linker fragments.

The overall focus of this study is the method to create the tri-pharmacophore libraries, which is well-executed. Nonetheless, the results from using this approach did not yield initial inhibitory compounds that showed distinct advantages over prior approaches.

Thank you for the comment. With the trio-pharmacophore format, we envision exploring the possibility of identifying linker moieties from selection since linker optimization remains a

challenge in dual-pharmacophore DEL technology. However, our current library needs expansion to yield more potent hits. The identified moieties can guide us in designing a full linker, e.g., by structure-based approaches.

T-DEL format can be applied as a follow-up step after identifying a potent fragment pair by a dual-pharmacophore library (Figures 2 and 3). In addition, the T-DEL format can also form a combinatorial library as large as a three-building block single-pharmacophore library without compromising the purity (Figures 4 and 5). Our current work represents the first step and a proof-of-principle to demonstrate the idea of self-assembling three sub-libraries. Further improvements to the library design, especially incorporating a code-joining mechanism (also discussed by the other two reviewers), would be needed in our future work. We have discussed these points in the revised manuscript.

REVIEWERS' COMMENTS

Reviewer #1 (Remarks to the Author):

In the revised version of the manuscript together and in their rebuttal letter Reddavid, Chen, Zhang and co-workers have indeed addressed all points that I had raised and that could still be included in this manuscript:

- The authors have now directly compared the de facto standard for stable dual-display libraries, ESAC technology, with the newly proposed triple pharmacophore setup, have included the respective results in the SI (SI pages 60f) and have added a respective paragraph in the main text. Doing so, the authors could reproduce previous ESAC-derived results by Wichert, M. et al. (2015) Nat Chem 7, 241–249 and Bigatti M, et al. (2017) ChemMedChem 2017 12(21), 1748-1752. They have now also included alpha-1-acid-glycoprotein (AGP) and the known pairs of binding fragments as qPCR-monitored selections and they have provided the respective docking results (Figure S10). Both dual- and triple pharmacophore setups gave the expected hits, while the triple pharmacophore results were "diluted" by the 30 DEL-B linkers. As assumed the DEL-B interactions with the DEL-A und DEL-C libraries yielded triple pharmacophores which are better and those which are worse than the dual-setup. This reflects the potential of the new approach and makes it interesting for fragment-based drug discovery. The authors also have found that the higher-affinity fragments of DEL-A,C yielded DEL-B linkers of higher enrichment. It will be interesting to see if this trend confirms in future triple-pharmacophore selections.

- My request to contrast the dual- and triple-pharmacophore results has also been fulfilled, with the result that same fragments from DEL-A and -C have been found as from the dual-display setup, which is reassuring.

- In my opinion, also the comments of the further reviewers have been addressed accordingly.

- As the authors themselves state, for de-novo selections without binding fragments at hand it will be important to develop a dedicated encoding scheme for the triple-pharmacophore libraries such that the three simultaneously binding fragments can be identified from the selections. I understand that this is a demanding task and that it is out of the scope of this proof-of-concept manuscript.

- In contrast to Reviewer 3 which finds the results for the MMP selections less compelling I think that with such small proof-of-concept DEL and given the difficulty to obtain non-hydroxamate high-affinity MMP ligands, it cannot be expected to yield new, high-affinity selective MMP hits. However, the obtained results show the potential of the method by identifying new lower-affinity fragments.

- The authors give all requested information except for the chemical building blocks of DEL-A and DEL-C, which is understandable since these DELs belongs to the company Dynabind.

Minor point:

On page 4 "Melkko, Neri, and co-workers postulated a triplex DEL in 200429,30 (Figure 1A)." should be given either as "Melkko, Scheuermann et al." or "Neri and co-workers".

In summary, the revised manuscript has fully addressed my requests. Since the new concept may prove useful for easing the typical dual-display related problem of finding a good linker between the two pharmacophores, I consider this manuscript very interesting to the whole DEL community and also all chemists interested in fragment-based drug discovery. As mentioned before, the manuscript as well as the SI are well written.

Hence, I recommend this work be published in Nature Communications.

Reviewer #2 (Remarks to the Author):

The authors have addressed all concerns and suggestions satisfactorily. Recommend to publish.

Reviewer #3 (Remarks to the Author):

In the revised manuscript the authors have done an excellent job in addressing the reviewers' comments and concerns. While fine-tuning is still needed, the present approach is innovative and holds much promise.

Reviewer 1

In the revised version of the manuscript together and in their rebuttal letter Reddavid, Chen, Zhang and co-workers have indeed addressed all points that I had raised and that could still be included in this manuscript:

- The authors have now directly compared the de facto standard for stable dual-display libraries, ESAC technology, with the newly proposed triple pharmacophore setup, have included the respective results in the SI (SI pages 60f) and have added a respective paragraph in the main text. Doing so, the authors could reproduce previous ESAC-derived results by Wichert, M. et al. (2015) Nat Chem 7, 241–249 and Bigatti M, et al. (2017) ChemMedChem 2017 12(21), 1748-1752. They have now also included alpha-1-acid-glycoprotein (AGP) and the known pairs of binding fragments as qPCR-monitored selections and they have provided the respective docking results (Figure S10). Both dual- and triple pharmacophore setups gave the expected hits, while the triple pharmacophore results were "diluted" by the 30 DEL-B linkers. As assumed the DEL-B interactions with the DEL-A und DEL-C libraries yielded triple pharmacophores which are better and those which are worse than the dual-setup. This reflects the potential of the new approach and makes it interesting for fragment-based drug discovery. The authors also have found that the higher-affinity fragments of DEL-A,C yielded DEL-B linkers of higher enrichment. It will be interesting to see if this trend confirms in future triple-pharmacophore selections.

- My request to contrast the dual- and triple-pharmacophore results has also been fulfilled, with the result that same fragments from DEL-A and -C have been found as from the dual-display setup, which is reassuring.

Thank you for the comment.

- In my opinion, also the comments of the further reviewers have been addressed accordingly. As the authors themselves state, for de-novo selections without binding fragments at hand it will be important to develop a dedicated encoding scheme for the triple-pharmacophore libraries such that the three simultaneously binding fragments can be identified from the selections. I understand that this is a demanding task and that it is out of the scope of this proof-of-concept manuscript.

Thank you very much for the comment. Incorporating a code-joining mechanism will accelerate the hit validation greatly. We will be working on this issue in due course.

- In contrast to Reviewer 3 which finds the results for the MMP selections less compelling I think that with such small proof-of-concept DEL and given the difficulty to obtain non-hydroxamate high-affinity MMP ligands, it cannot be expected to yield new, high-affinity selective MMP hits. However, the obtained results show the potential of the method by identifying new lower-affinity fragments.

Thank you for the comment. Our selections against MMP yielded compounds of micromolar affinity and the compounds do not contain typical strong zinc binders. The affinity increased after fragment linking, while the compounds remain to be improved by further optimization.

- The authors give all requested information except for the chemical building blocks of DEL-A and DEL-C, which is understandable since these DELs belongs to the company Dynabind.

Thank you very much for your understanding. Although we are not able to publish the structure of the compounds in SL-A and SL-C, the libraries are available commercially. The details are given in the method section.

Minor point:

On page 4 "Melkko, Neri, and co-workers postulated a triplex DEL in 200429,30 (Figure 1A)." should be given either as "Melkko, Scheuermann et al." or "Neri and co-workers".

We have edited the phrase as reviewer corrected.

In summary, the revised manuscript has fully addressed my requests. Since the new concept may prove useful for easing the typical dual-display related problem of finding a good linker between the two pharmacophores, I consider this manuscript very interesting to the whole DEL community and also all chemists interested in fragment-based drug discovery. As mentioned before, the manuscript as well as the SI are well written.

Hence, I recommend this work be published in Nature Communications.

Thank you very much for your comment.

Reviewer #2 (Remarks to the Author):

The authors have addressed all concerns and suggestions satisfactorily. Recommend to publish.

Thank you very much for the comment.

Reviewer #3 (Remarks to the Author):

In the revised manuscript the authors have done an excellent job in addressing the reviewers' comments and concerns. While fine-tuning is still needed, the present approach is innovative and holds much promise.

Thank you very much for the comment. We agree that more fine-tuning is needed in current T-DEL. In particular, we will investigate the code-joining mechanism to take full advantage of DEL as an FBDD approach.